# The separation pin distinguishes the pro- and anti-recombinogenic functions of *Saccharomyces cerevisiae* Srs2

Aviv Meir[1,5], Vivek B. Raina[1,5], Carly E. Rivera[1], Léa Marie [2,4], Lorraine S. Symington [2,3] & Eric C. Greene [1] ✉

Srs2 is an Sf1a helicase that helps maintain genome stability in *Saccharomyces cerevisiae* through its ability to regulate homologous recombination. Srs2 downregulates HR by stripping Rad51 from single-stranded DNA, and Srs2 is also thought to promote synthesis-dependent strand annealing by unwinding D-loops. However, it has not been possible to evaluate the relative contributions of these two distinct activities to any aspect of recombination. Here, we used a structure-based approach to design an Srs2 separation-of-function mutant that can dismantle Rad51–ssDNA filaments but is incapable of disrupting D-loops, allowing us to assess the relative contributions of these pro- and anti-recombinogenic functions. We show that this separation-of-function mutant phenocopies wild-type *SRS2* in vivo, suggesting that the ability of Srs2 to remove Rad51 from ssDNA is its primary role during HR.

Helicases use the energy derived from ATP hydrolysis to translocate along DNA or RNA substrates allowing them to fulfill a diverse range of functions, including the unwinding of duplex or structured nucleic acids and the remodeling or disruption of nucleoprotein complexes[1–10]. There are nearly 100 helicases encoded within the human genome, several of which play crucial roles in regulating homologous recombination (HR) and mutations in these helicases can give rise to severe genetic diseases characterized by genome instability and cancer predisposition[3,4,11–17].

Helicases can be divided into six super families, termed Sf1 through Sf6, which are defined by the amino acid sequence identity of their conserved helicase motifs[18–22]. Superfamily 1 (Sf1) is one of the largest and most diverse group of helicases and can be subdivided into two groups based on the direction of translocation: Sf1a helicases move in the 3′ → 5′ direction relative to the bound strand of nucleic acid and Sf1b helicases move in the opposite direction[4,6,7,10]. The Sf1 helicase core is comprised of four globular domains (1 A, 2 A, 1B and 2B), which together resemble a pair of tandem RecA-like folds with a single ATP-binding pocket residing in the center between domains 1 A and 2 A[23–26]. This core domain couples ATP binding and hydrolysis to protein conformational changes that enable helicases to move along nucleic acids[4,6,7,9,10,27]. The Sf1 helicase core contains at least seven conserved amino acid motifs (termed motifs Q, I, Ia, II, III, IV, V and VI), which form a bipartite ATP-binding pocket and a large portion of the nucleic acid-binding cleft[7,18,28–30]. Helicase core domains also contain a structural element referred to as "separation pins" or "wedges", which help facilitate nucleic acid unwinding[31–34]. For the Sf1a helicase family, the separation pin is found within domain 2 A and is positioned at the ssDNA/dsDNA junction to assist with strand separation[24].

The *S. cerevisiae* protein Srs2 is an Sf1a family member that has served as a paradigm for understanding helicase-mediated regulation of HR[6,35,36]. HR is an essential DNA repair pathway that allows for the exchange of genetic information between two different DNA molecules of identical or nearly identical sequence composition and is essential for the maintenance of genome integrity[37–39]. Srs2 is considered a proto-typical "antirecombinase" due to its well-characterized

---

[1]Department of Biochemistry & Molecular Biophysics, Columbia University, New York, NY 10032, USA. [2]Department of Microbiology & Immunology, Columbia University, New York, NY 10032, USA. [3]Department of Genetics & Development, Columbia University, New York, NY 10032, USA. [4]Present address: Institute of Pharmacology and Structural Biology (IPBS), French National Centre for Scientific Research (CNRS), Université Toulouse III, Toulouse, France. [5]These authors contributed equally: Aviv Meir, Vivek B. Raina. ✉e-mail: ecg2108@cumc.columbia.edu

ability to remove the recombinase Rad51 from ssDNA, resulting in a downregulation of HR[35,6,40–48]. Srs2 is also considered to have pro-recombinogenic functions where it acts by disrupting partially extended D-loops, which can then be directed towards repair through synthesis-dependent strand annealing (SDSA)[49–56]. In addition, Srs2 plays roles in the removal of mis-incorporated ribonucleotides from DNA[57], assists in unwinding triplet repeat hairpins during DNA replication[58–62], and removes RPA from ssDNA to dampen checkpoint signaling[47,63]. Human homologues of Srs2 have yet to be identified, although FBH1 and PARI are potential candidates[64,65]. A growing body of evidence suggests similar roles in HR might be filled by other human helicases, including RECQ1, RECQ5, BLM (Sgs1 in yeast), FANCM (Mph1 in yeast), FANCJ and RTEL1[1,4,,66]. Although *S. cerevisiae* Srs2 has served as a paradigm for understanding mechanistic aspects of HR regulation by helicases, structure-function studies of Srs2 itself have been hindered in part because no high-resolution Srs2 structure is available.

Here, we use a predicted AlphaFold model of Srs2 to help define and characterize amino acid residues that contact DNA. From this work, we have identified several amino acid residues that when mutated lead to compromised activity in bulk biochemical assays and single molecule biophysical assays, including examples of single amino acid residue changes that abolish the ability of Srs2 to translocate on Rad51- or RPA-bound ssDNA. Interestingly, our data also show that relatively subtle changes in in vitro biochemical and biophysical characteristics can lead to surprisingly strong genetic phenotypes, suggesting that the activity of Srs2 must be precisely tuned to assure physiologically relevant outcomes. In addition, we show that mutation of a highly conserved "separation pin" amino acid residue abolishes in vitro Srs2 helicase and D-loop disruption activity. However, this pin mutant retains wild-type levels of ssDNA-dependent ATP hydrolysis activity and can readily dismantle Rad51–ssDNA filaments in vitro, thus serving as a separation-of-function mutant that retains anti-recombinase activity but losses the capacity to unwind D-loops. Remarkably, we find that the *srs2* pin mutant phenocopies wild-type *SRS2* in a range of genetic assays, including assays for template switching, spontaneous recombination and DSB-induced crossover formation. These findings suggest that the most important biological attribute of Srs2 with respect to HR is its ability to physically remove Rad51 from ssDNA intermediates.

## Results

### Identification of potential Srs2 ssDNA contacts

Srs2 (1174 amino acids; 134 kDa) is a homolog of bacterial UvrD and both proteins fulfill similar roles in genome maintenance, thus structural and mechanistic studies of *E. coli* UvrD have direct bearing upon our understanding of Srs2[67,68]. Indeed, Srs2 and UvrD share 28% sequence identity and 40% similarity across their full lengths and 30% sequence identity and 80% similarity across their core helicase domains. Analysis of Srs2 structure and function relationships have been hindered due to the lack of high-resolution structural information. As an initial step towards helping to overcome this problem, we obtained a 3D model of Srs2 from AlphaFold[69,70] and compared the resulting model to the crystal structure of UvrD bound to a DNA fragment[31] (Fig. 1a–c). Analysis of the Srs2 model revealed a strong structural similarity to UvrD, as anticipated, yielding an RMSD of 0.374 Å across the core domain (Supplementary Fig. 1a–e).

Using the AlphaFold model as a guide, in conjunction with the structure of UvrD[31], we sought to identify key conserved amino acid residues in Srs2 that might contact ssDNA and therefore affect its ability to remove Rad51 from ssDNA, unwind dsDNA, or both (Fig. 1d, e). These amino acid residues include phenylalanine 68 (F68), asparagine 70 (N70), tyrosine 283 (Y283), phenylalanine 285 (F285) and arginine 286 (R286) in domain 1 A; histidine 100 (H100) and phenylalanine 219 (F219) in domain 1B; and arginine 389 (R389), histidine 650 (H650) and proline 671 (P671) in domain 2 A (Fig. 1d, e, Supplementary Fig. 1f). To test the

importance of each amino acid residue, we generated Srs2 mutant proteins in which each aforementioned residue was changed to alanine. In addition, we also generated a mutant protein in which tyrosine 775 (Y775) within domain 2 A was changed to alanine (Fig. 1f). Y775 corresponds to the separation pin, which is thought to play a role in unwinding duplex nucleic acids[31,32,34]. All mutants were made using GFP-tagged versions of Srs2, so that the recombinant proteins could be analyzed in single molecule DNA curtain assays (see below). This N-terminal GFP-Srs2 fusion construct retains biological functional in vivo[71] and retains biochemical activity in vitro[46,47]. In addition, full-length Srs2 has a strong tendency to aggregate, so GFP–Srs2 contains Srs2 amino acid residues 1–898 and the remaining 276 amino acid residues of the C-terminus were omitted. This truncated version of Srs2 retains wild-type levels of ATPase, DNA helicase, and Rad51 filament disruption activities[42,43,72]. For brevity, we will refer to GFP-Srs2 1-898 as Srs2[898] with respect to the in vitro assays.

### ATP hydrolysis activity of Srs2[898] point mutants

Srs2 exhibits robust DNA-dependent ATP hydrolysis activity in vitro[40,41,73]. The Srs2 mutant K41A has a lysine to alanine substation within the Walker A nucleotide-binding motif and is deficient for ATP hydrolysis activity[40]. ATP hydrolysis is intimately coupled to Srs2 function and as such the K41A mutant has no detectable helicase activity and cannot dismantle Rad51–ssDNA filaments[40,43,46]. All of the Srs2[898] mutants were expressed in *E. coli* and purified to near homogeneity (Supplementary Fig. 2a). ATP hydrolysis assays were conducted using 40 nM of each specified Srs2[898] protein and 0.5 to 8.0 mM ATP (as indicated). Reaction products were resolved by thin layer chromatography (Supplementary Fig. 2b), data quantitated using phosphor imaging and graphed as Michaelis-Menten plots for analysis (Fig. 2a). From this analysis, Srs2[898] yielded $K_M$, $V_{max}$, and $k_{cat}$ values of $1.58 \pm 0.71$ mM, $2.02\,\mu M/sec$ and $51 \pm 8\,s^{-1}$ in the presence of ssDNA, respectively (Fig. 2a, Supplementary Table 1). The most profoundly affected mutant proteins were F285A and H650A, which exhibited $K_M$ values that were 74% and 84% higher than WT, the $V_{max}$ values were reduced by 49% and 71%, and the $k_{cat}$ values were reduced by 51% and 71%, respectively (Fig. 2a, Supplementary Tables 1, 2). The F68A, P671A, and Y775A mutants were all largely unperturbed with respect to ATP hydrolysis activity, yielding kinetic parameters that were to within 4% of the Srs2[898] values (Fig. 2a, Supplementary Tables 1, 2). Whereas the N70A, H100A, F219A, Y283A, R268A, and R389A mutant proteins were all moderately affected, yielding kinetic parameters for ATP hydrolysis that were to within 18% of the Srs2[898] values. Taken together, these results indicate that amino acid residues F285 and H650 are important for the ssDNA–dependent ATP hydrolysis activity of Srs2[898]; F68, P671, and Y775 are not important for ATP hydrolysis; and the remaining amino acid residues (N70, H100, F219, Y283, R268, and R389) make more moderate contributions to Srs2[898] ATP hydrolysis activity.

### All Srs2[898] point mutants are proficient for ssDNA binding

The binding activity of the Srs2[898] point mutants in the absence of ATP was assessed by electrophoretic mobility shift assays (EMSA) using a 5′ fluorescein–labeled 40 nt ssDNA oligonucleotide substrate (Fig. 2b, Supplementary Fig. 2c). The fraction of bound substrate was quantified using a phosphor imager and equilibrium dissociation constants ($K_d$) were determined from the resulting binding curves (Fig. 2b, Supplementary Table 1). Examples of EMSA binding data are shown for Srs2[898] and for Srs2[898]–H650A, which was the point mutant that had displayed the largest reduction in ssDNA binding affinity (Supplementary Fig. 2c). These data revealed that Srs2[898]–H650A exhibited a 59% reduction in binding affinity compared to Srs2[898], yielding $K_d$ values of $46 \pm 3$ nM and $29 \pm 3$ nM, respectively (Supplementary Tables 1, 2). In addition, F285A, R389A, and P671A, also showed significant defects in ssDNA binding with $K_d$ values that were reduced by 41%, 24% and 31% with respect to WT Srs2 (Fig. 2b, Supplementary Tables 1, 2).

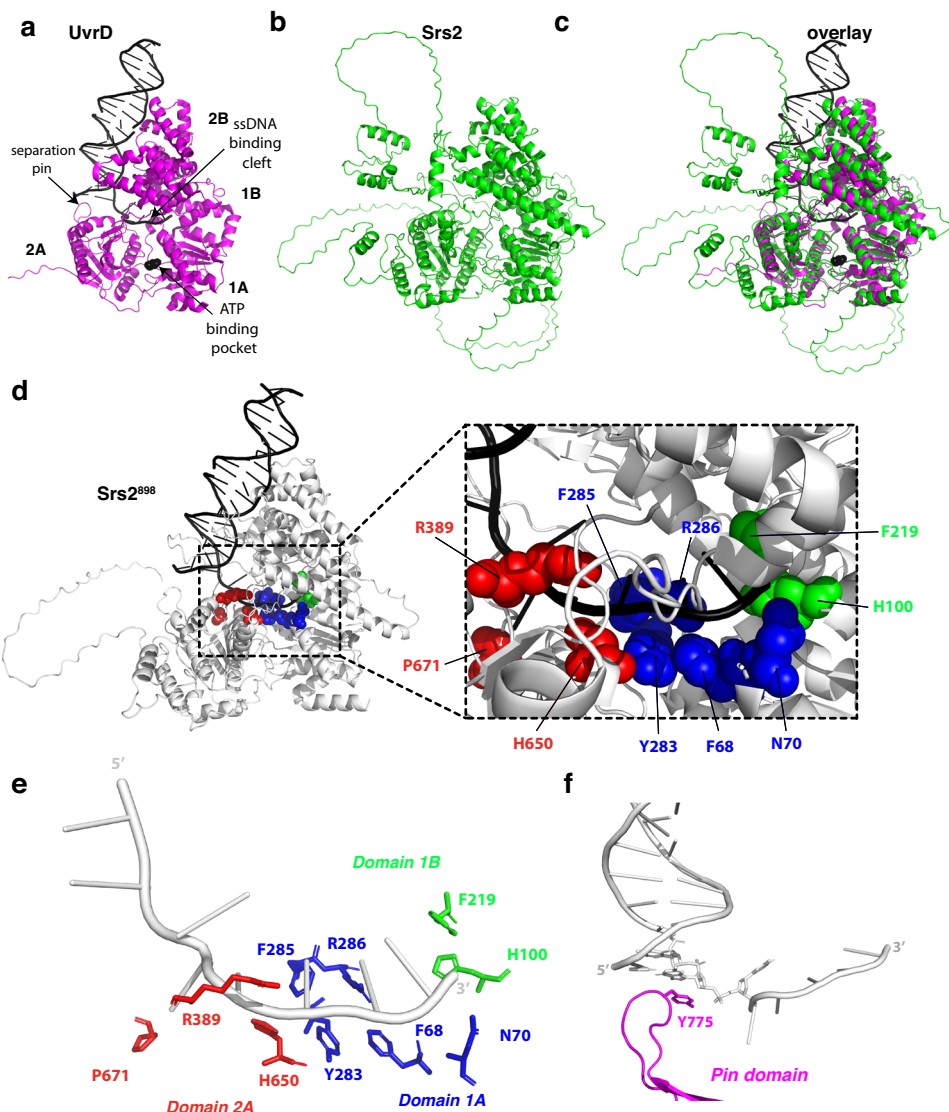

**Fig. 1 | Homology model of *S. cerevisiae* Srs2. a** The crystal structure of UvrD with bound DNA and ATP (PDB ID 2IS1)[31]. **b** Predicted AlphaFold model for *S. cerevisiae* Srs2[70,71]. **c** Merged structure of UvrD and the Srs2 AlphaFold model. **d** AlphaFold model of Srs2[898] superimposed with the DNA molecule from the UvrD structure. The Srs2 amino acid side chains predicted to be in close contact with the bound ssDNA are shown as color–coded space filling residues. **e** The Srs2 amino acid side chains predicted to be in close contact with bound ssDNA shown in the absence of the protein backbone. **f** Structure of the UvrD pin bound in complex with DNA (PDB ID 2IS1); the highlighted tyrosine residue is numbered according to the *S. cerevisiae* Srs2 amino acid sequence. Also see Supplementary Fig. 1.

In contrast to the five aforementioned proteins, most of the mutants did not exhibit appreciable changes in binding affinity compared to Srs2[898]; indeed F68A, N70A, H100A, F219A, Y238A, R268A, Y775A all yielded binding affinities that were to within ten percent of the value obtained for Srs2[898] (Fig. 2b, Supplementary Tables 1, 2). Notably, none of the mutations completely abolished ssDNA binding activity and even the point mutant that showed the greatest defect in binding affinity (H650A) still displayed what could be considered high binding affinity for ssDNA. So, although the reduction in binding affinity for the binding defective mutants displayed Kd values that were 24–59% higher than to Srs2[898], all of these proteins still bound ssDNA with reasonably high affinities, with the Kd values ranging from 36 to 46 nanomolar (Supplementary Tables 1, 2). The finding that the proteins could all bind tightly to ssDNA suggested that they were properly folded. As a further verification we measured the CD spectra for all of the Srs2[898] proteins used in this study and in each case, the mutant Srs2[898] proteins showed comparable CD spectra and comparable alpha helical content in comparison to Srs2[898] (Supplementary Fig. 3).

## Helicase activity of Srs2[898] mutants

The helicase activity of the Srs2 mutant proteins was measured using a Alexa 647–labeled dsDNA substrate with a 40–nt 3' overhang. All helicase assays were conducted in the presence of 2 mM ATP and an ATP regenerating system. The deproteinized reaction products were resolved by electrophoresis on 6% polyacrylamide gels (Supplementary Fig. 2d) and the fraction of unwound substrate was quantitated by phosphor imaging (Fig. 2c). Remarkably, the Y775A mutant, corresponding to an alanine substitution at a tyrosine residue within the pin domain, which is presumed to be necessary for dsDNA unwinding activity, resulted in a 97% reduction in the amount of DNA product unwound at the 30–minute time point (Fig. 2c, Supplementary Tables 1, 2). This finding highlights the importance of tyrosine 775 for the dsDNA unwinding activity of Srs2[898]. Srs2[898] bearing the F68, N70A and R286A exhibited helicase activity levels that were comparable to Srs2[898] (Fig. 2c, Supplementary Tables 1, 2). The H100A, F219A, Y283A, R389A, and P671A mutants were more compromised for helicase activity, revealing 18% to 45% decreases in helicase activity compared to Srs2[898] (Fig. 2c, Supplementary Tables 1, 2). Finally, the F285A and

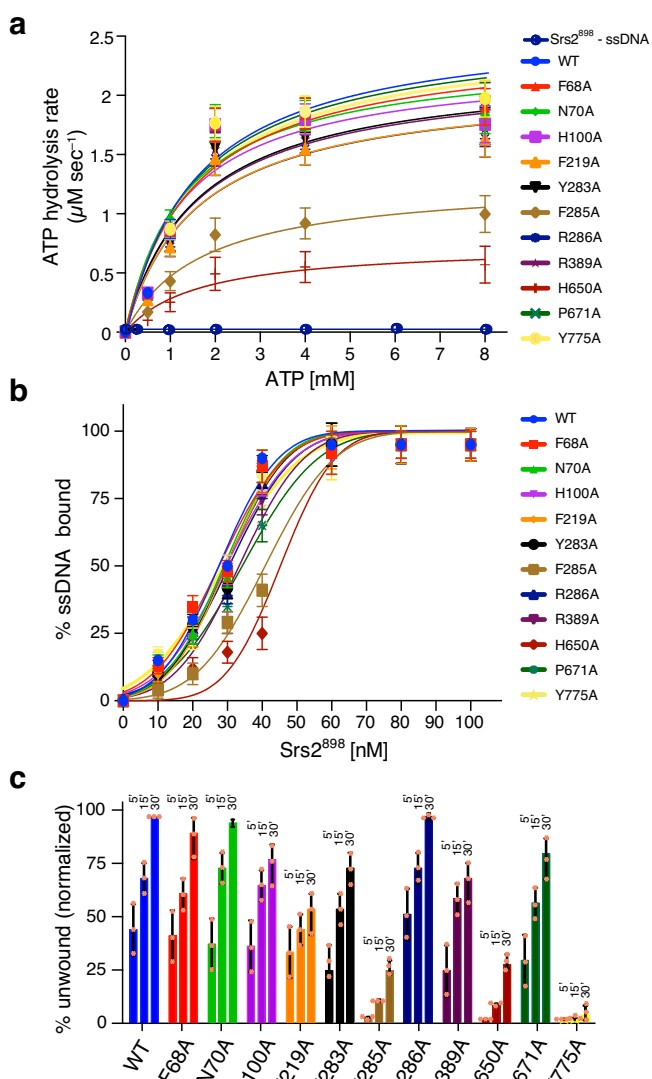

**Fig. 2 | The Srs2[898]-Y775A pin domain mutant is deficient for DNA helicase activity. a** Quantification of ATP hydrolysis rates for Srs2 mutants at varying concentration of ATP in the presence of ssDNA; error bars represent 95% CI; $n = 3$ independent experiments per data set. **b** Quantification of ssDNA binding activity for Srs2[898] mutants using a 40-nt ssDNA substrate; error bars represent 95% CI; $n = 3$ independent experiments per data set. **c** Quantification of DNA helicase assays for Srs2[898] mutants (40 nM each) using a substrate with a 40-bp dsDNA region flanked by a 40-nt 3′ ssDNA overhang. Successive bars within each color-coded data set represent 5-, 15- and 30-min time points for the reactions. Bar heights indicate the mean of three separate reactions ($n = 3$). Helicase reactions were performed for 5, 15 and 30 minutes, as indicated, and the 30 minute time point for Srs2[898] was used for normalization. In (**a**) through (**c**), all error bars represent standard deviation (SD) and all reactions were performed in triplicate. Values for all quantified reaction parameters are presented in Supplementary Table 1 and relative comparisons are presented Supplementary Table 2 and Supplementary Fig. 2 & 3.

H650A mutants were both highly compromised for helicase activity, exhibiting 75% and 72% reductions in the amount of unwound reaction products compared to Srs2[898] (Fig. 2c, Supplementary Tables 1, 2).

### Single molecule studies of Srs2[898] mutants

Srs2 can actively translocate along ssDNA while stripping both RPA and Rad51 from the ssDNA[40,41,43], and we have established single molecule ssDNA curtains assays allowing for the direct observation of GFP-tagged Srs2[898] in real time using total internal reflection fluorescence microscopy (TIRFM)[46,47,74]. In brief, long ssDNA substrates (≥ 50

kilonucleotides, knts) are generated by rolling circle replication using a 5′ biotin-labeled ssDNA primer and the resulting 5′ biotinylated ssDNA is tethered to a supported lipid bilayer on the surface of a microfluidic sample chamber through a biotin-streptavidin linkage[74]. The ssDNA molecules are then aligned at chromium (Cr) nanofabricated barriers to lipid diffusion, which are deposited onto the fused silica by electron beam lithography[74]. Addition of mCherry-labeled RPA allows the ssDNA to be extended by hydrodynamic force, the 3′ ends of the RPA-ssDNA become anchored to Cr pedestals through nonspecific adsorption, allowing the molecules to be visualized by TIRFM (Fig. 3a, b)[74]. Once assembled, the RPA can be displaced by the addition of Rad51 plus ATP resulting in the formation of long Rad51-ssDNA filaments[74]. Using these types of assays, we have previously reported that GFP-Srs2[898] translocates at a rate of $142 \pm 77$ nucleotides per second (nts/sec) for an average distance of $18.5 \pm 0.65$ kilonucleotides (knt) on ssDNA that is bound by Rad51 and $170 \pm 80$ nt/sec for an average distance of $14.4 \pm 0.40$ knt on RPA bound ssDNA[46,47].

Consistent with our previously published results, Srs2[898] exhibited translocation velocity and processivity values of $146 \pm 50$ nt/sec and $19 \pm 8.8$ knt, respectively, on Rad51-ssDNA[46]. The Srs2[898] mutants N70A, R286A, Y775A exhibited translocation velocity and processivity values that were not significantly different from Srs2[898] (Fig. 3c, d, Supplementary Tables 2, 3). The F68A mutant exhibited a moderate reduction in translocation velocity (16%) and a moderate reduction in processivity (17%) on Rad51-ssDNA compared to Srs2[898] (Fig. 3c, d, Supplementary Tables 2, 3). H100A, F219A, Y283A, R389A and P671A all exhibited large reductions, ranging from 35% to 50%, in translocation velocity on Rad51-ssDNA (Fig. 3c, d, Supplementary Tables 2, 3), and the F219A, R389A and P671A mutants also displayed correspondingly large reductions in processivity on Rad51-ssDNA (Fig. 3c, d, Supplementary Tables 2, 3). Notably, despite their reduced translocation velocity, the H100A and Y283A mutants still translocated for long distances on Rad51-ssDNA comparable to Srs2[898] (Fig. 3c, d, Supplementary Tables 2 & 3). Finally, the Srs2 mutations F285A or H650A exhibited no translocation activity on Rad51 bound ssDNA (Fig. 3c, Fig. 3d, Supplementary Tables 2 & 3).

Srs2[898] exhibited translocation velocity and processivity values of $179 \pm 61$ nt/sec and $15.4 \pm 4.3$ knts, respectively, on RPA-ssDNA, and these results were again consistent with our previously published findings[47]. Like the findings for Rad51-ssDNA, the Srs2[898] mutants N70A, R286A, Y775A exhibited translocation velocity and processivity values that were not significantly different from Srs2[898] on RPA-ssDNA (Fig. 3e, Fig. 3f, Supplementary Tables 2 & 3). The R286A mutant displays a moderate 23% reduction in velocity and a 16% reduction in processivity on RPA-ssDNA, although these values did not differ significantly from Srs2[898], and it also behaved similarly to Srs2[898] in assays with Rad51-ssDNA (Fig. 3e, f, Supplementary Tables 2, 3). While the F68A mutant exhibited a 34% reduction in velocity and 17% reduction in processivity on RPA-ssDNA (Fig. 3e, f, Supplementary Tables 2, 3), the H100A, F219A, Y283A, R389A and P671A mutants were more deficient for translocation on RPA-ssDNA and displayed reductions in velocity ranging from 51% to 61% of that observed for Srs2[898], and displayed similarly large decreases in processivity, ranging from 44–64% (Fig. 3e, f, Supplementary Tables 2, 3). Finally, as with Rad51-ssDNA, Srs2[898] bearing the F285A or H650A mutations exhibited no evidence of translocation activity on RPA-ssDNA (Fig. 3e, f, Supplementary Table 2, 3).

### Growth phenotypes of Srs2 mutants in the presence of MMS
*S. cerevisiae* cells exposed to the DNA alkylating agent methyl methane sulfonate (MMS) suffer DNA damage that is channeled through a Rad6- and Rad18-dependent translesion synthesis (TLS) repair pathway[72,75,76]. The Rad6-Rad18 ubiquitin E3 ligase complex targets PCNA for mono-ubiquitination, thus enabling translesion polymerases to bypass MMS-induced damage (Fig. 4a)[6,72,75–77]. It is thought that Srs2

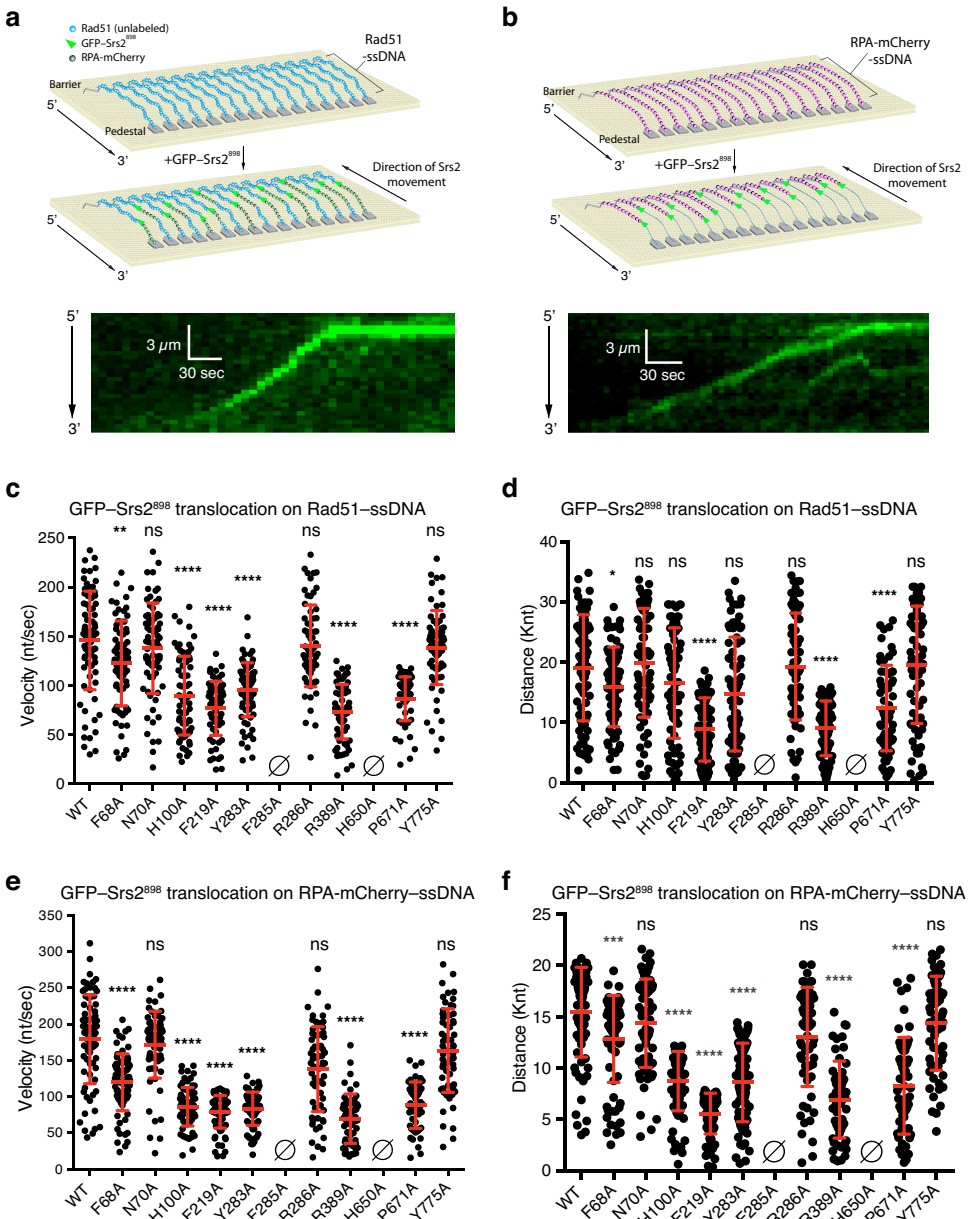

**Fig. 3 | Srs2[898]–Y775A can remove Rad51 and RPA from ssDNA. a** DNA curtain assay schematic for Srs2[898] translocation activity on Rad51–ssDNA (top) and an example kymograph showing Srs2[898] movement along a Rad51–ssDNA molecule (bottom). **b** Assay schematic for Srs2[898] translocation activity on mCherry-tagged RPA–ssDNA (top) and a kymograph showing Srs2[898] movement along an mCherry-tagged RPA–ssDNA molecule (bottom). Note that Srs2 moves in the 3′ to 5′ direction on the ssDNA as indicated by the arrows in the schematic diagrams. Quantification of Srs2[898] translocation (**c**) velocity (nt/sec) and (**d**) distance (knt) on Rad51 bound ssDNA. Srs2[898] translocation (**e**) velocity (nt/sec) and (**f**) distance (knt) on RPA bound ssDNA. Error bars in (**c**) through (**f**) represent SD. All translocation parameters are available in Supplementary Table 3, and relative comparisons are presented Supplementary Table 2.

helps channel MMS damage into the TLS pathway by hindering the formation of Rad51–ssDNA filaments at ssDNA gaps caused by MMS damage, thus downregulating HR (Fig. 4a)[6,72,75,76]. Deletion of the *RAD18* gene disrupts TLS, resulting in reduced cell growth on media containing MMS[72,75]. However, this *rad18Δ* growth defect is alleviated by deletion of *SRS2*, which allows MMS–induced DNA damage to be efficiently redirected through the Rad51–dependent template switching (TS) pathway in *rad18Δ srs2Δ* cells[72,75,78]. Thus, cell growth on media containing MMS provides a means of assessing the functional status of antirecombinase activity in vivo for each of the *srs2* mutants in vivo (Fig. 4a)[51,72,75,76].

Control experiments confirmed that deletion of *RAD18* inhibited cell growth on YPD plates in the presence of 0.005% MMS, and this

growth inhibition was relieved upon deletion of the *SRS2* gene (Supplementary Fig. 4a). Full length wild type (WT) *SRS2*, along with 500 bp upstream of the start codon corresponding to its promoter sequence (*p500-SRS2*), was cloned in an integrative vector and integrated at the *HIS3* genomic locus. Reintroduction of *p500-SRS2* at the *HIS3* locus in the *rad18Δ srs2Δ* background resulted in a loss of cell growth on media containing 0.005% MMS showing that *p500-SRS2* at the *HIS3* locus behaves similar to *SRS2* at the endogenous locus. In contrast, a control strain where an empty vector was integrated at the *HIS3* locus exhibited no growth defect (Supplementary Fig. 4a). The *srs2–N70A, –R286A*, and *–Y775A* mutants behaved similarly to WT *SRS2* and failed to alleviate the rad18Δ growth defect in the presence of MMS, indicating that these three mutants were capable of downregulating HR

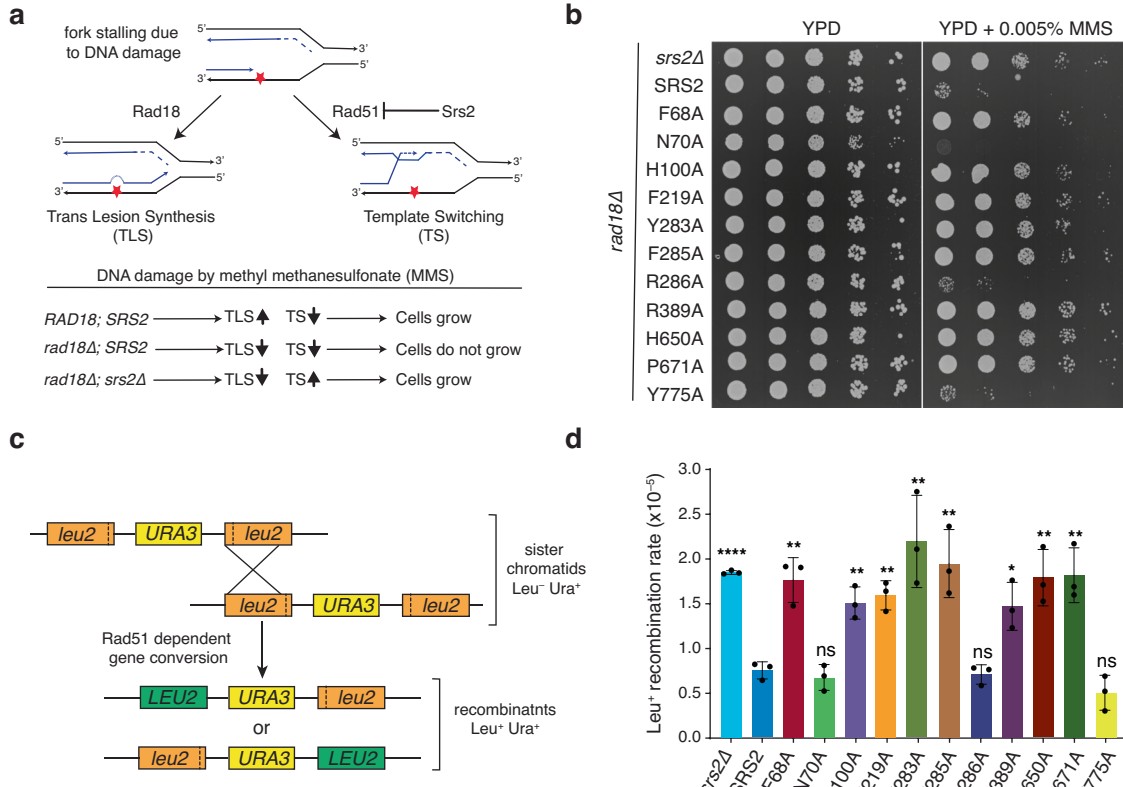

**Fig. 4 | Yeast expressing full-length Srs2-Y775A phenocopy wild-type strains.** **a** Schematic illustration of expected outcomes of methyl methanesulfonate (MMS) treatment in wild type, *rad18Δ*, and *rad18Δ srs2Δ* strains. **b** Serially diluted haploid yeast strains harboring the indicated point mutations on either YPD plates or YPD containing 0.005% MMS, as indicted. **c** Schematic of the direct repeat assay for spontaneous recombination via Rad51-dependent gene conversion between *leu2* heteroalleles. **d** Graph showing the spontaneous recombination rate determined from three independent fluctuation tests performed on the indicated strains. Error bars represent the standard deviations from the mean. Asterisks represent statistical significance determined by unpaired two-tailed *t* tests with exact p-values from left to right being <0.0001, 0.0028, 0.4752, 0.0030, 0.0015, 0.0089, 0.0062, 0.6245, 0.0121, 0.0054, 0.0046, 0.1173; ns (not significant); Also see Supplementary Fig. 4.

(Fig. 4b). In striking contrast, all remaining *srs2* mutants displayed robust cell growth in the *rad18Δ* background when grown on media containing 0.005% MMS (Fig. 4b). These results imply that none of these mutants downregulated HR to an extent sufficient to alleviate the *rad18Δ* growth defect on MMS plates.

Given that this genetic assay reflects the antirecombinase activity of Srs2, it is useful to consider the functional characteristics of the mutant Srs2$^{898}$ protein in the DNA curtain assays. Notably, N70A, R286A and Y775A all behave most similarly to Srs2$^{898}$ on Rad51-ssDNA filaments in the DNA curtain assays (Supplementary Table 2). The findings that *srs2-F285A* and *srs2-H650A* were not functional in vivo was not surprising, given that both mutants were incapable of removing Rad51 from ssDNA in the DNA curtain assays (Fig. 3c, d, Supplementary Tables 2, 3). In addition, the Srs2$^{898}$ mutants H100A, F219A, Y283A, R389A, and P671A all had more severe defects in Rad51 filament disruption with reductions in velocity and processivity ranging from 39–50% and 13–54%, respectively, (Fig. 3c, d, Supplementary Tables 2, 3), consistent with their inability to function as effective antirecombinases in vivo (Fig. 4b). However, the F68A mutant had only moderate defects of Rad51 filament disruption in vitro with reductions in velocity and processivity of just 16% and 17%, respectively (Fig. 3c, d, Supplementary Table 2) and yet it was still not able to function as a fully effective antirecombinase in the MMS assays (Fig. 4b). This latter result suggests that even moderate deficiencies in Srs2 antirecombinase activity levels are not well tolerated in the *rad18Δ* background in response to MMS-induced DNA damage. Importantly, the finding that *srs2-Y775A* yields results similar to WT *SRS2* indicates that

dsDNA unwinding activity is not necessary for *SRS2*-mediated downregulation of HR in the *rad18Δ* background in response to MMS-induced DNA damage.

**Hyper-recombination assays**

Defects in *SRS2* can give rise to a hyper-recombination phenotype that can be quantitatively assessed using a direct repeat recombination assay that reports on spontaneous Rad51-dependent intrachromosomal and inter-sister recombination (Fig. 4c)[72,79]. Therefore, we next asked whether cells expressing the *srs2* mutants exhibited a hyper-recombination phenotype. In this assay, the reporter construct contains two mutant *leu2* alleles separated by a functional *URA3* marker and spontaneous recombination between the two mutant *leu2* alleles can give rise to Leu$^+$ colonies which can grow on media lacking leucine. Consistent with prior studies[51,72,80], loss of *SRS2* caused a hyper-recombination phenotype reflected as an approximately 4-fold increase in the Leu$^+$ recombination rate (Fig. 4d), and control experiments confirmed that this phenotype could be rescued by addition of *p500-SRS2* but not with an empty vector (Supplementary Fig. 4b). Cells expressing the *srs2* mutant alleles *F68A, H100A, F219A, Y283A, F285A, R389A, H650A*, and *P671A* all exhibited hyper-recombination phenotypes comparable to the *srs2Δ* strain, suggesting a loss of Srs2 antirecombinase function in these cells (Fig. 4d). In contrast, cells expressing the *srs2* mutant alleles *N70A* and *R286A* were all similar to WT *SRS2* (Fig. 4d).

Surprisingly, cells expressing the helicase deficient *srs2-Y775A* allele also exhibited a Leu$^+$ recombination rate that was indistinguishable from WT *SRS2* (Fig. 4d). The finding that *srs2-Y775A* behaves similarly to WT

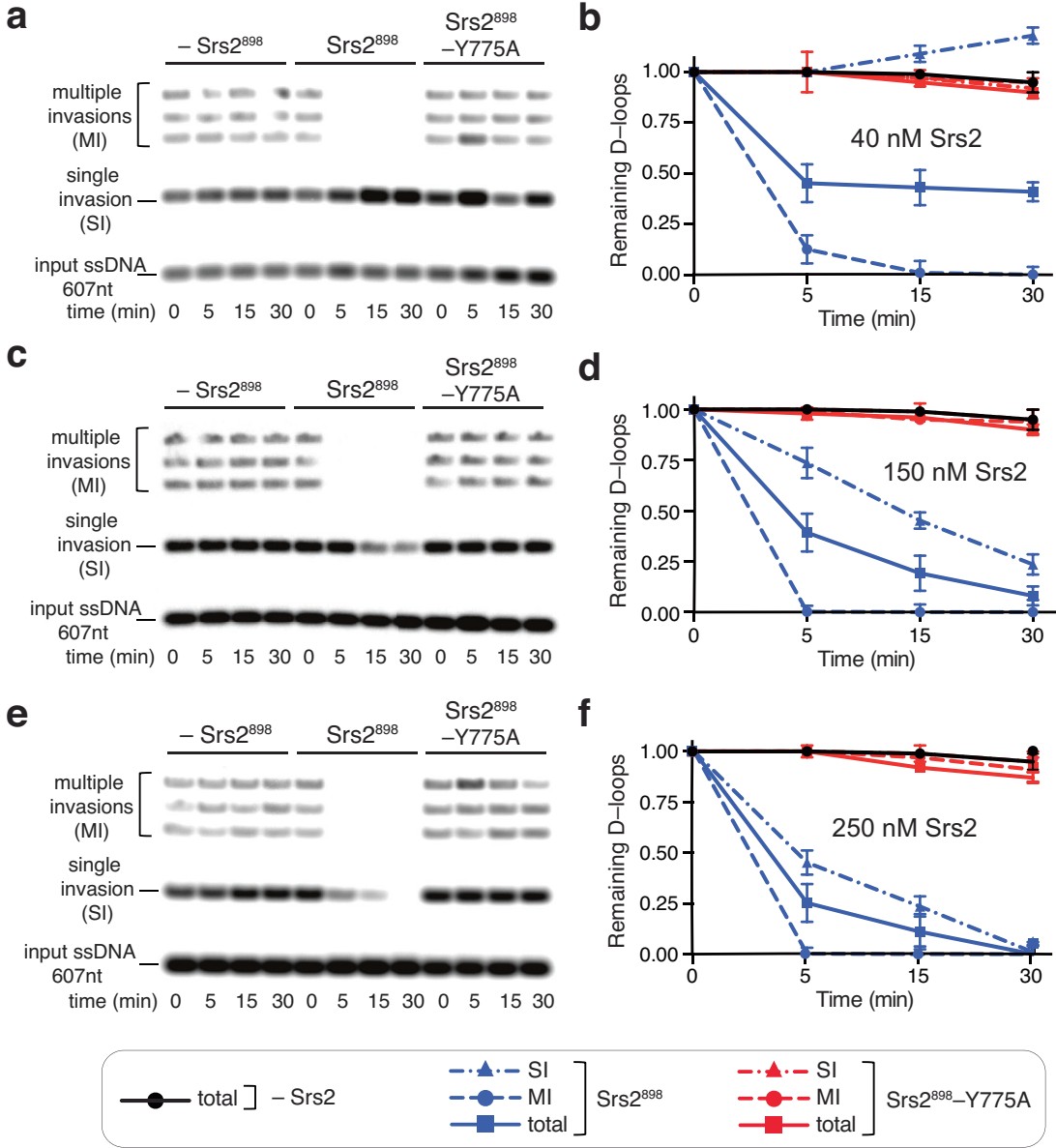

**Fig. 5 | Srs2<sup>898</sup>–Y775A is unable to disrupt Rad51-mediated D-loops.**
**a**, **c**, **e** Gel-based assays for D-loop disruption by Srs2. In all assays, D-loops were prepared using a fluorescently labeled 607-nt ssDNA fragment and a supercoiled plasmid (pUC19; 2,686 bp) in reactions with Rad51 and Rad54. Reactions were then incubated with the indicated concentration of Srs2<sup>898</sup> (40, 150 or 250 nM) and terminated before resolving on an agarose gel ($n = 3$). **b**, **d**, **f** Quantitation of the D-loop products at each indicated concentration of Srs2<sup>898</sup> ($n = 3$). All reactions were performed in triplicate; error bars represent SD.

*SRS2* suggests that dsDNA unwinding activity is not necessary for Srs2-mediated downregulation of spontaneous Rad51-dependent recombination in this direct repeat recombination assay.

**The Srs2<sup>898</sup>–Y775A pin domain mutant fails to disrupt D-loops**
The finding that *srs2-Y775A* did not exhibit a hyper-recombination phenotype in the direct repeat assay was unexpected because it is thought that the ability of Srs2 to disrupt D-loops is a key facet of its ability to suppress hyper-recombination. However, given that Srs2<sup>898</sup>–Y775A is severely compromised for in vitro helicase activity but retains the ability to disrupt Rad51 filaments at wild-type levels, we considered it likely that this mutant would be defective for D-loop disruption. To verify this hypothesis, we tested whether the Srs2<sup>898</sup>–Y775A pin domain mutant protein was able to disrupt extended D-loop intermediates generated in vitro in reactions with Rad51 and Rad54, as previously described[51]. In these assays, Rad51 filaments were assembled in the presence of 2.5 mM ATP onto an

ATTO647-labeled ssDNA substrate 607-nt in length that was homologous to the plasmid pUC19, followed by the addition of RPA[51]. Rad54 and supercoiled pUC19 were then added to generate the D-loop products reflecting the invasion of a single pUC19 plasmid (single invasion, SI) as well as the simultaneous invasion of up to four plasmids (multiple invasions, MI; Fig. 5)[51]. D-loop disruption was then initiated by the addition of 40, 150 or 250 nM Srs2<sup>898</sup> or Srs2<sup>898</sup>–Y775A, as indicated, and aliquots of the reaction mixture were sampled at 5-, 15-, and 30-minute time points (Fig. 5). Negative control reactions with no added Srs2<sup>898</sup> (–Srs2<sup>898</sup>) confirmed that the D-loop intermediates did not spontaneously dissociate in the absence of Srs2<sup>898</sup> (Fig. 5). As expected, the D-loop products were rapidly disrupted by the addition of Srs2<sup>898</sup> (Fig. 5), consistent with previous reports[51]. In striking contrast, the Srs2<sup>898</sup>–Y775A pin domain mutant was unable to disrupt the D-loop products and instead yielded outcomes comparable to the negative control reactions lacking Srs2 (Fig. 5). Taken together, our findings show that the Srs2-Y775A pin domain mutant retains the

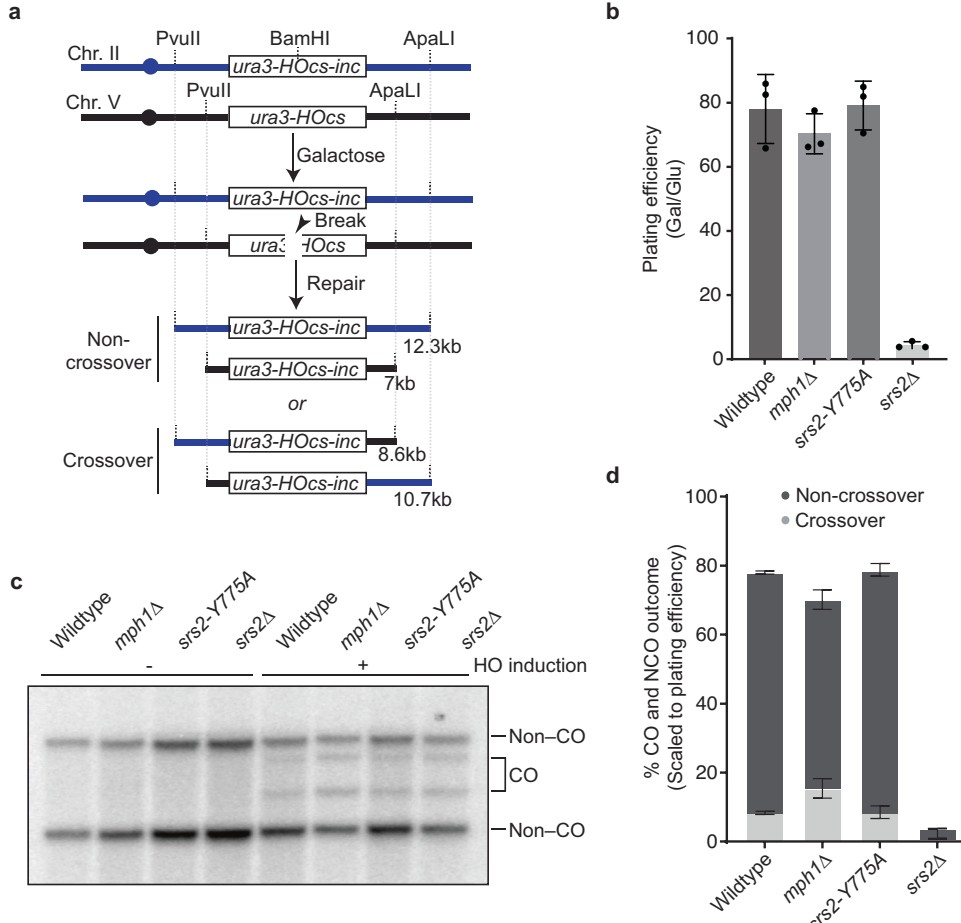

**Fig. 6 | Loss of D–loop disruption activity does not alter CO outcomes between ectopic repeats. a** Schematic of the assay used to detect CO and NCO recombinants formed between ectopic *ura3* repeats consisting of cleavable HO locus (*ura3–HOcs*) or non-cleavable HO locus (*ura3–HOcs–nc*) bearing a BamHI restriction site. **b** Plating efficiency of the strains with indicated genotype after HO induction as compared to the uninduced control (*n* = 3 biological repeats). Error

bars represent the standard deviations from the mean. **c** Southern blot analysis of DNA extracted from cells of the indicated genotype with or without HO induction. **d** Percent distribution of the recombinants shown for the indicated strains scaled to the plating efficiency (*n* = 3 biological repeats). Error bars represent the standard deviations from the mean.

ability to remove Rad51 filaments from ssDNA at a level comparable to WT Srs2 but is severely compromised in its ability to unwind dsDNA and as a consequence cannot disrupt D–loops.

**Physical analysis of ectopic recombination in *srs2–Y775A* cells**
In ectopic recombination assays where DSBs are induced by the HO endonuclease, *srs2Δ* cells exhibit a significant reduction in the formation of non–crossover products, whereas the level of crossovers remains unaffected[54,56]. These findings have led to the hypothesis that Srs2 suppresses crossover formation by disrupting D–loop intermediates and instead promotes the SDSA pathway which leads to non–crossover recombination products[51]. This pro-recombinogenic function of Srs2 can be assessed in vivo by measuring the relative ratio of crossover (CO) to non–crossover (NCO) outcomes in assays that use DSB–induced recombination between ectopic chromosomal repeats[81–84]. In this assay, the strains have an HO endonuclease cut site (*HOcs*) inserted within the native *ura3* locus on chromosome (Ch) V and a donor cassette, consisting of the non-cleavable HO cut site (*HOcs-inc*), containing 5.6 kb of the *ura3* region inserted in the *LYS2* locus on Ch II (Fig. 6a). The expression of HO is controlled by the galactose (*pGAL1*) promoter. HO endonuclease is responsible for initiating gene conversion at the mating type (*MAT*) locus by inducing DSBs. To circumvent this issue, the *MAT**a*** site on Ch III is modified to *MAT**a**-inc* allele which is refractory to HO digest and thus prevents

cleavage of the *MAT* locus. Addition of galactose to the growth media enables expression of the HO endonuclease that generates a DSB at the *HOcs* site, while the *HOcs-inc* remains refractory to its action. The presence of a unique BamHI site within the *HOcs-inc* donor cassette is used to monitor repair by gene conversion. NCO and CO products can be distinguished based upon the lengths of DNA fragments produced by digestion with PvuII and ApaLI (Fig. 6a).

Our data show that the Srs2[898]–Y775A mutant protein readily removes Rad51 from ssDNA but is severely compromised for helicase activity and cannot disrupt extended D–loops generated by Rad51 and Rad54. Therefore, we asked whether the ratio of CO to NCO recombination products was altered in cells expressing the *srs2-Y775A* mutant allele. Deletion of *MPH1* leads to a higher CO to NCO ratio and was thus used as a positive control[81]. BamHI digestion of the DNA isolated from cells exposed to galactose–containing media confirmed that all strains underwent HO–mediated gene conversion (Supplementary Fig. 5a). Note that the plating efficiency of the *srs2Δ* was very poor, with just 4.1 ± 1.1% of the cells surviving galactose–induced expression of HO endonuclease compared to 78.1 ± 10.8% and 70.4 ± 6.3% of the WT *SRS2* and *mph1Δ* cells surviving, respectively, after HO induction. (Fig. 6b). The growth defect observed for the *srs2Δ* strain was consistent with previous reports, which showed that *SRS2* was necessary for recovery from a Mec1-dependent checkpoint mitotic arrest arising after the induction of double stranded DNA

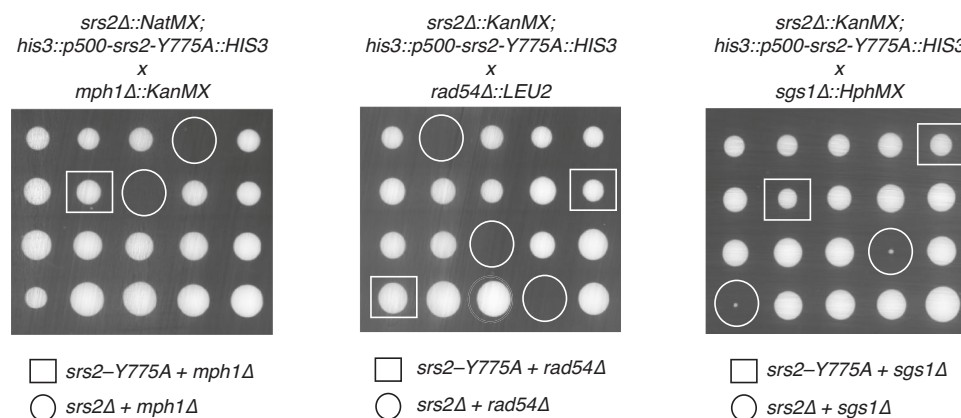

**Fig. 7 | Tetrad dissections of yeast strains heterozygous for *srs2-Y775A* and either *mph1Δ*, *rad54Δ*, or *sgs1Δ*.** The squares surround viable spores harboring *SRS2-Y775A* and either *mph1Δ, rad54Δ, or sgs1Δ*, as indicated. The circles indicate unviable or sick spores corresponding to the double mutant of *srs2Δ* and either *mph1Δ, rad54Δ, or sgs1Δ*. Spore clones from each tetrad are shown vertically.

breaks[85]. Interestingly, this cell cycle arrest phenotype was rescued in the strain expressing *srs2-Y775A* yielding a plating efficiency of 79.2 ± 7.63% (Fig. 6b), suggesting that Srs2 helicase and D-loop disruption activities are not required for this checkpoint recovery.

Consistent with prior studies, the *mph1Δ* strain showed an increase in the fraction of CO products compared to the WT *SRS2* strain (Fig. 6c, d, Supplementary Fig. 5b, c)[81]. The *srs2Δ* and *mph1Δ* strains both showed similar increases in the fraction of CO products compared to WT *SRS2* (Fig. 6c, d, Supplementary Fig. 5b, c), potentially implicating Srs2 in the regulation of CO and NCO reaction outcomes. However, considering that the plating efficiency of *srs2Δ* cells was substantially lower than the *mph1Δ* cells, these data must be interpreted with caution because the results do not imply that Srs2 and Mph1 behave equivalently with respect to CO/NCO regulation. Indeed, upon scaling the CO and NCO levels to the plating efficiency, the CO and NCO events in *srs2Δ* were observed to be significantly lower than in the *mph1Δ* cells (Fig. 6d). Most importantly, the *srs2-Y775A* strain yielded CO outcomes that were essentially indistinguishable from WT *SRS2* (Fig. 6c, d, Supplementary Fig. 5b, c). Taken together, these findings indicate that the roles of Srs2 in enabling checkpoint recovery after DSB formation and in regulating crossover outcomes does not arise from a need to disrupt D-loop intermediates.

## Synthetic lethal analysis of *srs2-Y775A*
Deletion of *SRS2* shows synthetic lethal interactions with mutation of *RAD54*, *MPH1*, and *SGS1*, all of which encode ATP-dependent DNA motor proteins that participate in homologous recombination[6,56,86–89]. We considered the possibility that one of these other helicases might act redundantly to accommodate the loss of DNA unwinding and D-loop disruption activity observed for srs2-Y775A, in which case it might be expected that *srs2-Y775A* should show synthetic lethal interactions with *rad54Δ*, *mph1Δ*, and *sgs1Δ*. As expected, based on previous studies[6,56,86–89], *srs2Δ* was synthetic lethal with *rad54Δ*, *mph1Δ* and *sgs1Δ* (Fig. 7). However, for all three cases, this synthetic lethality was rescued by *srs2-Y775A* (Fig. 7). These observations suggest that Rad54, Mph1 or Sgs1 are not acting in the place of Srs2-Y775A when it is incapable of disrupting D-loops, and instead further suggests that the D-loop disruption activity of Srs2 is less important than its ability to strip proteins such as Rad51 and RPA from ssDNA with respect to HR.

## Discussion
Through our analysis of Srs2 amino acid residues thought to make important contacts with DNA, we identified an Srs2 separation-of-function mutant, Srs2-Y775A, that is defective for D-loop disruption activity but still retains wild-type levels of

antirecombinase activity in vitro. We then used this mutant to characterize the relative contributions of Srs2 antirecombinase and D-loop disruption activities to homologous recombination in vivo. Our findings suggest that the D-loop disruption activity of Srs2 observed in vitro does not contribute significantly to the ability of Srs2 to regulate HR in vivo. Instead, our work suggests that the phenotype observed for *srs2Δ* strains arise from the inability to dismantle Rad51 filaments, rather than from any defect in D-loop disruption.

## The pin domain distinguishes Srs2 pro- and anti-recombinogenic functions
Helicase pin (or "wedge") domains are highly conserved structural features that are required for efficient separation of duplex nucleic acid strands[31–34]. For example, in the cases of UvrD and PcrA, tyrosine 621 and phenylalanine 626, respectively, stack against the first base pair of dsDNA, which is presumed to stabilize DNA unwinding intermediates thus enabling these proteins to unwind nucleic acids[24,31]. Alignment of UvrD and PcrA with Srs2 indicate that the tyrosine at position 775 serves as the amino acid residue significant for pin domain function[31]. Interestingly, the loss of helicase activity for the Srs2[898] pin domain (motif VIa) mutant Y775A (97% reduction in activity; Supplementary Table 2) is more profound than was observed for the equivalent mutants of UvrD (Y621 A; ~40% reduction in activity [31]) and PcrA (F626A; 25% reduction in activity[90]), suggesting that strand separation by Srs2 is much more reliant upon this tyrosine residue than either of the bacterial homologs. However, there was no a priori reason to believe that the pin domain should be necessary for Srs2[898] to remove Rad51 from ssDNA given that this functional attribute should not require any strand separation activity. Consistent with these considerations, the Y775A mutation disrupts helicase and D-loop disruption activities, but the DNA binding and ATP hydrolysis activities remain unaffected. Moreover, Srs2[898]-Y775A remains fully functional for translocation on both RPA-ssDNA and Rad51-ssDNA. Given that Srs2[898]-Y775A retained wild-type levels of the activities necessary for its antirecombinase functions, but lacked D-loop disruption activity, we considered the possibility that this protein could be used as a separation-of-function mutant that would only be defective for normal helicase-dependent biological functions. Surprisingly, cells expressing *srs2-Y775A* phenocopied WT *SRS2* in assays for: (1) cell growth on MMS-containing media in a *rad18Δ* background (Fig. 4a, b); (2) spontaneous recombination between direct repeats (Fig. 4c, d); (3) cell cycle arrest in response to a HO-induced DSB (Fig. 6b); and (4) the regulation of crossover and non-crossover outcomes in an ectopic recombination assay (Fig. 6c, d). In addition, we find that *srs2-Y775A* rescues the synthetic lethality normally observed between *srs2Δ* and

*rad54Δ*, *mph1Δ*, and *sgs1Δ* (Fig. 7). Taken together, the outcomes of these genetic assays strongly suggest that the loss of in vitro D-loop disruption activity for *srs2-Y775A* does not greatly affect its in vivo functions with respect to the aforementioned HR assays.

As indicated above, the Y775A pin domain amino acid residue is highly conserved among the Sf1a helicases, strongly suggesting that it must contribute to an important functional role for Srs2. Our results raise the question of what purpose Srs2 helicase activity might serve with respect to its biological functions. Given that Srs2[898]–Y775A is defective for helicase activity, we speculate that this mutant may have potential problems in preventing replication errors at triplet repeat sequences[61,62] and would likely be incapable of assisting with the removal of misincorporated ribonucleotides from DNA[57]. There may also exist other unidentified functions for the Srs2 helicase activity (independent of its functions in regulating HR) and the *srs2-Y775A* mutant may prove useful for testing this possibility.

### Structural and functional analysis of Srs2

Our analysis of Srs2 amino acid residues implicated as potential components of the ssDNA binding pocket reveal a range of biochemical defects when mutated to alanine. The Srs2[898] mutants F68A, N70A, R286A, all of which reside within domain 1A, were only moderately affected in their bulk biochemical activities, with greatest impact observed for F68A, which exhibited a 3.9% reduction in its $k_{cat}$ for ATP hydrolysis, a 14% reduction in helicase activity, a 16% reduction in its translocation velocity on Rad51−ssDNA and a 34% reduction in its translocation velocity on RPA−ssDNA (Supplementary Table 2). The Srs2[898] mutants H100A, F219A, Y283A, R389A, P671A were more strongly affected, highlighting the importance of these amino acid residues for Srs2 biochemical activities. Interestingly, there appeared to be a general trend where defects in ATP hydrolysis and ssDNA binding affinity seem to be magnified in the helicase and single molecule translocation assays with RPA− and Rad51−bound ssDNA. For instance, the H100A mutant shows a 12% reduction in its $k_{cat}$ for ATP hydrolysis and a 39% and 52% reduction in translocation velocity on Rad51−ssDNA and RPA−ssDNA, respectively (Supplementary Table 2). One possible implication of these observations is that defects in ssDNA translocation may become more apparent when Srs2 must exert force while displacing a strand of DNA or while acting against ssDNA-bound proteins. Interestingly, for most of the mutants, the magnitude of the translocation defects was almost always greater in the case of RPA−ssDNA compared to Rad51−ssDNA, suggesting that deficiencies in Srs2[898] translocation may have a greater relative impact upon its ability to remove RPA from ssDNA as compared to Rad51, perhaps because RPA has a much higher affinity for ssDNA compared to Rad51[91–93].

The Srs2[898] mutants F285A and H650A each have relatively high ssDNA binding affinities and they still retain the ability to hydrolyze ATP albeit in both cases they are reduced relative to Srs2[898], however, they are both more profoundly deficient for helicase activity (Supplementary Table 2). Moreover, neither of these two mutants shows any evidence for translocation activity on either RPA−ssDNA or Rad51−ssDNA (Supplementary Table 2). Taken together, these findings strongly suggest that the F285A and H650A mutants have lost functional motor protein activity. Interestingly, amino acid residue W256 in UvrD (F285 in Srs2) has been designated as an "anchor" amino acid because it remains immobile during the ATP hydrolysis cycle[31,32]. We speculate that mutation of this tryptophan amino acid to alanine may result in a mutant protein that is incapable of coupling the protein structural transitions that normally take place during the ATP binding and hydrolysis cycle with the binding transitions that must take place at the protein−DNA interface, thus losing the capacity to actively translocate on ssDNA. Given that Srs2 amino acid residues F285 and H650 reside close to one another in 3D space, near the middle of the ssDNA binding pocket (Fig. 1d, e), it is possible that H650A acts

similarly to decouple ATP binding and hydrolysis from protein movement. Interestingly, the PcrA mutant W259A behaves somewhat differently than the equivalent mutation in Srs2 (F285A). PcrA W259A exhibits ATP hydrolysis behavior comparable to WT but had a 200-fold decrease in ssDNA binding activity and a 300-fold decrease in DNA helicase activity[90]. In comparison, Srs2[898]–F285A has 1.4-fold (41% lower) reduction in ssDNA binding, ~50% reduction in $V_{max}$ and $k_{cat}$ for ATP hydrolysis and 91% reduction in helicase activity (Supplementary Tables 1, 2). In these cases, the ultimate outcome is the same for the PcrA-W259A and Srs2[898]-F285A mutations − a defective motor protein incapable of unwinding DNA or presumably translocation on ssDNA − but these effects manifest in distinct ways. This difference suggests that highly conserved amino acid residues may play subtly different roles in different but closely related helicases.

### Defects in Srs2 translocation activity can have a big impact upon in vivo function

Interestingly, for many of the mutant Srs2 proteins seemingly moderate defects in their in vitro activities have big effects on phenotype. For example, Srs2[898]–F68A, H100A, F219A, Y283A, R389A and P671A all exhibited reduced in vitro activities compared to Srs2[898] (Supplementary Table 2). For instance, these proteins all had reduced translocation velocities on Rad51−ssDNA ranging from a 16.8% reduction for Srs2−F68A (122 ± 43 nt/sec) to a 50% reduction for Srs2[898]−R389A (73 ± 27 nt/sec) compared to Srs2[898]. In each case, these mutants phenocopied an *srs2Δ* strain with respect to MMS sensitivity in a *rad18Δ* background (Fig. 4b) and exhibited hyper-recombination phenotypes that closely resembled a *srs2Δ* strain (Fig. 4d). These findings are similar to previous observations for an Srs2 truncation mutant lacking 314 amino acids from the C−terminus (Srs2 1−860), which is unable to function as an effective antirecombinase in vivo[72] but was able to translocate at a velocity of 77 nt/sec on Rad51−ssDNA[46]. These results suggest that moderate Srs2 translocation defects are not well tolerated, likely reflecting a delicate balance between the need to assemble and dismantle Rad51 filaments, as well as a need to disrupt ssDNA-bound RPA to dampen checkpoint signaling, thus allowing for proper HR regulation.

### Conclusion

Our findings suggest that the main function of Srs2 with respect to the regulation of HR appears to be its ability to remove Rad51 and RPA from ssDNA. The design principles used here to disrupt helicase and D−loop disruption activities of Srs2, while leaving its motor protein and antirecombinase functions intact, may allow for the construction of similar separation−of−function mutants for other helicases that play regulatory roles in recombination such as the BLM and RECQ5 helicases.

### Methods

#### Homology modeling

The *S. cerevisiae* Srs2 AlphaFold model (Uniprot code: P12954) was uploaded to PyMOL 2.4.1, (Schrödinger) and aligned with the structure of UvrD (PDB 2IS1)[31]. RMSD was calculated in Pymol. Potential Srs2 ssDNA contacts were defined by superimposing the Srs2 structure model with the crystal structure of UvrD (PDB 2IS1)[31].

#### Mutagenesis

Srs2[898] point mutations were generated using an In-Fusion® HD Cloning kit (Takara Bio Inc., Cat. No. 102518). The DNA sequences encoding the point mutations were amplified by PCR using the gene for GFP-tagged Srs2[898] as a template[46,47]. The primers used for amplification were then digested by DpnI (New England Biolabs; Cat. No. R0176S) at 37 °C for 15 min. The PCR products were ligated using 5X In-Fusion HD Enzyme Premix (Takara Bio Inc.; Cat. No. 102518), transformed into Stellar cells (Takara Bio Inc.; Cat. No. 636766) and then plated onto LB-

agar plates (supplemented with 100 µg/mL carbenicillin) and grown at 37 °C. Single colonies were then re-grown overnight in 10 mL LB (supplemented with 100 µg/mL carbenicillin) at 37 °C. Plasmid DNA was purified using a miniprep purification kit (Promega; Cat. No. A1460). All mutant plasmids were verified by sequencing (Genewiz).

## Proteins

*S. cerevisiae* RPA and Rad51 were expressed and purified according to previously described method with slight modifications as stated below[94]. mCherry-RPA (6xHis-tagged) was expressed in *E. coli* pLysS cells. A single colony was inoculated into 6 L of LB containing 50 µg/ml carbenicillin and induced at 0.9 OD 600 using 0.5 mM IPTG. Cells were grown overnight at 16 °C and then harvested by centrifugation at 4000× g for 25 min at 4 °C. The cell pellet was resuspended in lysis buffer (500 mM NaCl, 20 mM Tris [pH 7.5], 2 mM β-ME, 5 mM imidazole, 10% glycerol, and 0.5 mM PMSF) and then lysed by sonication. The lysate was clarified by centrifugation at 25 000 rpm for 30 min at 4 °C, and the clarified lysate was bound to Ni-NTA resin (Qiagen) equilibrated in R-buffer (50 mM KCl, 20 mM Tris−HCl [pH 7.4], 1 mM dithiothreitol (DTT), 0.5 mM ethylenediaminetetraacetic acid (EDTA), 10% glycerol) for 1 h in batch at 4 °C on a rotator. The Ni-NTA resin was then washed with R-buffer plus 50 mM imidazole and mCherry-RPA was eluted using R-buffer with 200 mM imidazole. The pooled fractions were dialyzed into R buffer and then fractionated on a heparin column. Protein was eluted from the heparin column using a linear gradient 50–1000 mM KCl in R buffer. Pooled fractions were dialyzed into R150 buffer (150 mM KCl, 20 mM Tris [pH7.4], 1 mM DTT, 0.5 mM EDTA, 50% glycerol) overnight at 4 °C using 10 000 MWCO Snakeskin™ dialysis tubing (Thermo-Fisher Scientific). Fractions were then flash frozen in liquid nitrogen and stored at −80 °C. Rad51 (6xHis-tagged) was expressed in *E. coli* Rosetta2 cells. An overnight bacterial culture was diluted 50-fold in 2xLB media supplemented with ampicillin (100 µg/ml) and grown at 37 °C to OD 600 = 0.6–0.8. Rad51 expression was induced with 0.1 mM IPTG for 3 hours at 37 °C. Cell lysate preparation and all the protein purification steps were conducted at 4 °C in buffer T (50 mM Tris-HCl, pH 7.5, 10% glycerol, 1000 mM NaCl, 15 mM imidazole, 1 mM EDTA, 0.01% IGEPAL CA-630 (Sigma), 1 mM DTT) supplemented with 2 mM ATP and 2 mM MgCl₂. Cells were disrupted by sonication. After ultracentrifugation (100,000×g for 90 min), the lysate was incubated with 2 ml of Talon affinity resin (Clontech) for 2 h with gentle mixing. The matrix was poured into a column with an internal diameter of 1 cm and washed sequentially with 20 ml of buffer with 2.5 mM Imidazole and with 15 mM KCl, respectively, followed by ScDmc1 elution using buffer supplemented with 150 mM Imidazole and eluted using 200 mM imidazole. The protein pool was diluted with an equal volume of buffer T and fractionated in a 1 ml Heparin Sepharose column (GE Healthcare) with a 30 ml gradient of 150–1000 mM KCl, collecting 1 ml fractions. Fractions containing Rad51 (eluting at ~500 mM KCl) were pooled, diluted to the conductivity of 150 mM KCl and further fractionated in a 1 ml Mono Q column with a 30 ml gradient of 150–500 mM KCl, collecting 1 ml fractions. Fractions containing Rad51 (eluting at ~300 mM KCl) were pooled, concentrated in an Amicon Ultra micro-concentrator (Millipore), snap-frozen in liquid nitrogen, and stored at −80 °C.

Srs2[898] and the Srs2[898] mutants were expressed and purified as previously described, with some minor modifications as described below[46,47]. *E. coli* Rosetta2 (DE3) cells (Novagen) were transformed with a pET15b vector containing GFP-tagged Srs2[898] (or mutant variants) and plated onto LB-agar plates supplemented with 100 µg/mL carbenicillin and 40 µg/mL chloramphenicol. Single colonies were then selected and grown at 37 °C overnight in 25 mL of LB 100 µg/mL carbenicillin and 40 µg/mL chloramphenicol. 2 mL of this culture was used to inoculate 2 L of LB medium supplemented with 200 µg/mL carbenicillin and 40 µg/mL chloramphenicol, and cultures were grown at 37 °C with continuous shaking to an OD₆₀₀ of ~1.0. The temperature

was then reduced to 16 °C and protein expression was initiated with the addition of 0.1 – 0.5 mM isopropyl-β-D-thiogalactopyranoside (IPTG). Cells were grown for an additional 20 h at 16 °C with slow shaking (80 RPM). Cells were then harvested by centrifugation at 4,000 rpm for 20 minutes, and the resulting cell pellet was frozen on liquid nitrogen and stored at −80 °C. The frozen cell pellet was thawed at 37 °C and resuspended in cell lysis buffer containing 40 mM NaHPO₄ [pH 7.5], 600 mM KCl, 5% glycerol, 10 mM imidazole [pH 7.8], 0.1 mM TCEP (Tris(2-carboxyethyl)phosphine hydrochloride; Sigma, Cat. No. C4706), 0.05% Tween-20, 10 µM E-64 (Sigma-Aldrich, Cat. No. E3132), 100 µM AEBSF (4-(2-aminoethyl)benzenesulfonyl fluoride hydrochloride; Sigma-Aldrich, Cat. No. A8456), 1 mM benzamidine, and 1 mM PMSF (phenylmethanesulfonyl fluoride; Sigma-Aldrich, Cat. No. P7626). Cells were lysed by sonication on ice and the resulting lysate was clarified by ultracentrifugation at 25,000 rpm for 45 minutes. The clarified lysate (~30–40 mL) was then incubated for 1 hour with a Talon metal affinity resin (5 mL bed volume per liter of culture; Clontech; Cat. No. 635503) equilibrated with Buffer A (40 mM NaHPO₄ [pH 7.5], 300 mM KCl, 5% glycerol, 15 mM imidazole, 0.02% Tween-20, 1 mM benzamidine, 1 mM PMSF, and 0.125% myoinositol). The column was then washed extensively with Buffer A. Srs2 was eluted from the Talon metal affinity column with a step of Buffer A plus 400 mM imidazole [pH = 7.8]. Immediately after elution, the sample was adjusted to 5 mM EDTA [pH 8] and 1 mM TCEP. The eluate was then dialyzed in Snake-Skin Dialysis Tubing (10,000 MWCO; Thermo Scientific; Cat. No. 68100) against 1 L of Heparin Buffer (20 mM NaHPO₄ [pH 7.5], 100 mM KCl, 5% glycerol, 0.01% Tween-20, 1 mM TCEP, 2 mM EDTA, 0.125% myoinositol) for 1.5 h, with 1 L buffer changes every 30 min. The dialyzed eluate was then loaded onto a 5 mL HiTrap Heparin column (GE Lifesciences; Cat. No. 17-0406-01) equilibrated with Heparin Buffer, and the proteins were eluted with a single step of Heparin Buffer containing 500 mM KCl. The peak fraction (~4 mL) was then dialyzed in SnakeSkin Tubing (10,000 MWCO) against storage buffer (40 mM NaHPO₄ [pH 7.5], 300 mM KCl, 10% glycerol, 0.01% Tween-20, 1 mM TCEP, 0.5 mM EDTA, 0.125% myoinositol) for 2 h at 4 °C. Protein concentrations were determined using the Quick Start Bradford Protein Assay Kit (Bio-Rad, Cat. No. 5000201). The samples were then aliquoted, flash frozen in liquid nitrogen, and stored at −80 °C.

## ATP hydrolysis assays

ATP hydrolysis assays were performed in Srs2 reaction buffer (30 mM Tris-Cl [pH 7.6], 100 mM KCl, 2 mM MgCl₂, 1 mM DTT, 0.2 mg mL⁻¹ BSA) in the presence of 0.5 to 8 mM cold ATP (as indicated) and trace amounts of γ−[³²P]ATP (3000 Ci/mmol; Perkin Elmer, Cat. No. BLU502A250UC). All reactions contained 2.5 µM (in nucleotides) of M13 ssDNA (NEB, Cat. No. N4040S), reactions were initiated by the addition of Srs2[898] to a final concentration of 40 nM and reactions were then incubated at 30 °C. Aliquots were removed at specified time points and quenched with the addition of 25 mM EDTA [pH 8.0] and 1% SDS. The quenched reactions (2 µl) were spotted on 20 ×20 cm TLC Silica gel plates (Sigma-Aldrich; Cat. No. HX02446579) and resolved in 0.5 M LiCl plus 1 M formic acid. Dried TLC plates were exposed to a phosphor imaging screen and scanned with a Typhoon FLA 9000 (GE Healthcare). All data were obtained from three separate experiments and were fit in PRISM (GraphPad). Rates of ATP hydrolysis in units of µM/sec were determined by quantitating the amount of ATP hydrolyzed as a function of time for each different ATP concentration and the resulting data were plotted as rate of ATP hydrolysis (µM/sec) versus ATP concentration [mM]. Kinetic parameters ($K_M$, $V_{max}$ and $k_{cat}$) were obtained by fitting the curves in Prism (version 6.0) using a nonlinear regression.

## DNA binding assays

Srs2[898] DNA binding assays were performed at 30 °C for 10 min in 10 µl Srs2 buffer (30 mM Tris-HCL [pH 7.6], 100 mM KCl, 2 mM MgCl₂, 1 mM

DTT, 0.2 mg mL$^{-1}$ BSA) and reactions contained increasing concentrations of Srs2$^{898}$ (10 nM-100 nM, as indicated) together with 1.3 μM (in nucleotides) of a 40-nt ssDNA fluorescein-labeled oligonucleotide (IDT) of the following sequence: 5′-Fluorescein-ATT AAG CTC TAA GCC ATG AAT TCA AAT GAC CTC TTA TCA A-3′. Reactions were mixed with 10 μl of Stop buffer (50% glycerol, 20 mm Tris-HCl [pH 7.4], 2 mm EDTA) and resolved in 10% native polyacrylamide gels in TAE buffer (40 mM Tris-acetate [pH 8], 1 mm EDTA) at 4 °C for 2 h. Gels were scanned at a wavelength of 473 nm using a Typhoon FLA 9000 (GE Healthcare). The fluorescently labeled DNA bands were quantified using the open-source software package Fiji[95] and the ratio between the bound fraction to the total DNA (bound fraction plus unbound fractions) was plotted. All data were obtained from three separate experiments and were fit in PRISM (GraphPad). K$_D$ values were extracted from the fitted curves and correspond to the concentration of Srs2$^{898}$ at which 50% of the ssDNA substrate bound.

## CD spectra

CD measurements were performed using a Chirascan spectrometer (Applied Photophysics Inc.) using protein samples diluted into buffer containing 100 mM NaF, 1 mM TCEP-HCl, 5 mM Tris [pH 7.5]. The final concentration of Srs2$^{898}$ for each measurement was 70 nM. Measurements were made at 25 °C using a cell path length of 0.5 mm. CD traces were obtained from 180 nm and 280 nm using a 1 nm step size and 1 nm bandwidth, and the traces were background corrected using a buffer blank (100 mM NaF, 1 mM TCEP-HCl, 5 mM Tris [pH 7.5]). Ten scans were collected for each protein and the average of these ten scans is shown in the graph. CD spectra were fitted and the percent of alpha helical content for the different Srs2$^{898}$ mutants were calculated[96].

## Helicase assays

The helicase assay conditions for the substrate with a 3′ ssDNA overhang were adapted from a previously published protocol[97]. The DNA substrate for the helicase assays was prepared by annealing 5′-ATT AAG CTC TAA GCC ATG AAT TCA AAT GAC CTC TTA TCA A- Alexa 647-3′ and 5′-TTG ATA AGA GGT CAT TTG AAT TCA TGG CTT AGA GCT AAA TTG CTG AAT CTG GTG CTG GGA TCC AAC ATG TTT TAA ATA TG-3′ in annealing buffer (100 mM Tris-HCl [pH 7.5], 500 mM NaCl, and 100 mM MgCl$_2$). The annealing mix is placed in a 95 °C water bath for 5 min, and the water bath was then placed on a benchtop and allowed to cool slowly to room temperature. The annealed product was then resolved on a 6% polyacrylamide gel by crush and soak and purified using a NucleoSpin Gel and PCR Clean-up kit (Macherey-Nagel, Cat. No. 740609). Assays were initiated by mixing 40 nM Srs2$^{898}$ in 10 μl of reaction buffer (30 mM Tris-HCl [pH 7.5], 2.5 mM MgCl$_2$, 2 mM ATP, 100 mM KCl, 1 mM dithiothreitol, and 100 μg/ml BSA) containing an ATP-regenerating system consisting of 20 mM creatine phosphate and 20 μg/ml creatine kinase and the DNA substrate (25 nM). Reactions were incubated at 30 °C and aliquots removed at the indicated times and terminated with addition of a 2× stop solution (0.4% SDS and 50 mM EDTA). Samples were then deproteinized by addition of 0.5 μl of proteinase K (0.5 mg/ml final) and then incubated at 37 °C for 15 min. The reaction products were resolved on a 6% nondenaturing polyacrylamide gels in TBE buffer (40 mM Tris-Borate, [pH 7.4], 0.5 mM EDTA) at 4 °C. Gels were scanned at a wavelength of 635 nm with a Typhoon FLA 9000 (GE Healthcare). The fluorescently labeled DNA bands were quantified using the open-source software package Fiji[95] and the percent of the unwound dsDNA substrate was calculated as the intensity of the ssDNA band divided by the total intensity of ssDNA and dsDNA multiplied by 100. The normalized percent of unwound DNA was plotted by setting the percent of Srs2$^{898}$ at 30 min to 100% and fitting the rest of the data accordingly. All data were obtained from three separate experiments and were fit in PRISM (GraphPad).

## D−loop disruption assays

D−loop disruption assays were performed essentially as previously described, with some modifications[51]. In brief, a fluorescently-tagged 607-nt ssDNA homologous to the plasmid pUC19 (NEB, Cat No. N3041A) was generated by PCR using the following primers (Integrated DNA Technologies): 5′-ATTO647-CGC GAG ACC CAC GCT CAC CGG CTC CAG ATT TAT CAG CAA TAA A-3′ and 5′-Phos-TGC ACG AGT GGG TTA CAT CGA ACT GGA TCT CAA CAG CGG TAA GA-3′. PCR reaction was performed using 0.2 μM of the purified oligos and 0.6 ng/ml pUC19 plasmid (NEB, Cat # N3041A) in CloneAmp HiFi PCR premix (TaKaRa, Cat #639298). PCR annealing was set for 63 °C with 3 min synthesis time. PCR products were purified on a 2% agarose gel followed by NucleoSpin Gel and PCR cleanup (MACHEREY-NAGEL, Cat #740609.10). The PCR product was treated with λ exonuclease (Bio-Labs, Cat #M0262S) to digest the non−fluorescent strand according to manufacturer protocol. In brief, the PCR product (90 ng/μl) was incubated with 1 μl (5 units) λ exonuclease (BioLabs, Cat #M0262S) in λ exonuclease buffer (BioLabs, Cat #M0262S), in a total volume was 50 μl. The reaction was incubated at 37 °C for 30 min followed by the addition of EDTA (10 mM, final concentration) and incubation at 75 °C for 10 min to inactivate the λ exonuclease. The resulting ATTO 647−labeled ssDNA was purified on a 2% agarose gel followed by NucleoSpin Gel and PCR cleanup (MACHEREY-NAGEL, Cat #740609.10). D−loop experiments were performed in HR buffer (30 mM Tris−OAc [pH 7.5], 50 mM KCl, 20 mM MgOAc, 1 mM DTT, 0.2 mg/ml BSA). The ATTO 647−labeled ssDNA substrate (10 nM) was incubated with 2 μM Rad51 for 10 min at 30 °C in HR buffer plus 2.5 mM ATP. RPA (300 nM) was then added to the reaction mix and the incubation continued for an additional 15 min. Rad54 (200 nM) and supercoiled pUC19 (NEB, Cat #N3041A) were then added to the reaction mix and the incubation continued for an additional 15 min to generate the D−loop products. Srs2$^{898}$ (40, 150 or 250 nM, as indicated) was then added to the D−loop reaction mix and incubated at 30 °C for 5, 10, 15 or 30 min, as indicated. Reactions were terminated with the addition of an equal volume of Stop buffer (20 mM EDTA, 1% SDS, and 20% glycerol). The reaction products were then de−proteinized by the addition of Proteinase K (0.5 mg/ml, final concentration), and then were resolved on a 0.9% agarose gel in 1xTAE buffer. Gels were scanned using a Typhoon FLA 9000 with a 635−nm laser to detect the ATTO 647 dye (GE Health Sciences) and DNA species were quantified using the open-source software package Fiji[95]. Graphs were generated by normalizing the data for reactions containing Srs2$^{898}$ relative to the minus Srs2 samples. The graphed data points represent the mean and error bars represent the SD of triplicate experiments.

## DNA curtain assays and data analysis

All experiments were conducted with a custom−built prism−type total internal reflection fluorescence (TIRF) microscope (Nikon) equipped with a 488−nm laser (Coherent Sapphire, 200 mW), a 561−nm laser (Coherent Sapphire, 200 mW), and two Andor iXon EMCCD cameras[74,98]. To prepare flowcells, chrome barriers were deposited on quartz microscope slides via e−beam lithography and thermal evaporation, as described[98,99]. In brief, lipid bilayers were prepared with 91.5% DOPC (1,2−dioleoyl−sn−glycero−3−phosphocoline), 0.5% biotinylated−PE (1,2−dioleoyl−sn−glycero−3−phosphoethanolamine−N−(cap biotinyl)), and 8% mPEG 2000−DOPE (1,2−dioleoyl−sn−glycero−3−phosphoethanoloamine−N−[methoxy (polyethylenegycol)−2000]) (Avanti Polar Lipids, Inc., Cat. No. 850375 P, 870273 P and 880130 P, respectively). Lipid bilayers were deposited in preformed flow chambers through sequential deposition of a lipid master mix in lipid buffer (20 mM Tris−Cl [pH 7.5], 100 mM NaCl).

The ssDNA substrate was generated by rolling circle replication using phi29 DNA polymerase with a biotinylated primer annealed to M13 circular single stranded DNA as a template[74,98]. The ssDNA was

tethered to the bilayer through a biotin–streptavidin linkage in BSA buffer (40 mM Tris-HCl [pH 8.0], 2 mM MgCl₂, 1 mM DTT, 0.2 mg/mL BSA)[74,98]. The ssDNA molecules were aligned at a flow rate of 0.5 ml/min in BSA buffer plus 0.1 nM RPA (or RPA–mCherry). Once ssDNA molecules were aligned, the flow rate was adjusted to 1.0 mL/min and 0.5 mL of 7 M urea was injected into the flow cell to further extend the ssDNA. BSA buffer (40 mM Tris-HCl [pH 8.0], 2 mM MgCl₂, 1 mM DTT, 0.2 mg/mL BSA) containing 0.1 nM RPA-mCherry was then flushed through the sample chamber for 8 to 10 min rate of 0.8 mL/min. Rad51 filament formation was initiated by injecting HR buffer (30 mM Tris-Ac [pH 7.5], 50 mM KCl, 5 mM MgAc, 1 mM DTT, 0.3 mg/mL BSA, and 2 mM ATP) and 1 mM Rad51, followed by a 15 min incubation in the absence of buffer flow. Flow was resumed with HR buffer (30 mM Tris-Ac [pH 7.5], 50 mM KCl, 5 mM MgAc, 1 mM DTT, 0.3 mg/mL BSA, and 2 mM ATP) at 0.5 mL/min for 5 min to flush out any remaining free Rad51. GFP-Srs2[898] (0.5 nM) was then injected through a 150 mL sample loop, and translocation activity was observed in the absence of buffer flow. All single molecule assays were conducted at 30 °C.

Image acquisition was initiated concurrently with 0.5 nM GFP-Srs2[898] injection in HR buffer (30 mM Tris-Ac [pH 7.5], 50 mM KCl, 5 mM MgAc, 1 mM DTT, 0.3 mg/mL BSA, and 2 mM ATP) at a frame rate of 1 frame per 10 s for a total time of approximately 20 min. Data was collected with a 100–millisecond integration time and the lasers were shuttered between images to minimizing photo–bleaching. Images were collected using Nikon software, and images were exported as individual TIFF images[74,98]. TIFF stacks were imported into ImageJ (Fiji)[95]. For two–color imaging, the two channels were first corrected for stage drift and then merged into TIFF images, which were then converted to TIFF stacks. All TIFF stacks were then corrected for stage drift using the registration/translation function within Fiji[74]. For each time course experiment, kymographs were generated from the TIFF image stacks by defining a 1–pixel wide region of interest (ROI) along the axis of each individual ssDNA molecule, and these ROIs were extracted from each image within the TIFF stack[74]. All slices corresponding to one ssDNA molecule were then aligned to yield a kymograph representing the entire experimental time course, and this process was repeated for each ssDNA molecule that was analyzed[74]. 1 pixel corresponds to -1087 nts of RPA-ssDNA, while for the Rad51-ssDNA 1 pixel corresponds to -725 nucleotides nts[46,47,74]. Velocities and distances traveled were calculated from the kymographs and values calculated as follows: $velocity = [(Y_f - Y_i) \times 1000 nt]/[(X_f - X_i) \times frame\ rate]$; $distance = (Y_f - Y_i) \times 1000 nt$; where $Y_i$ and $Y_f$ correspond to the initial and final positions of Srs2 along the ssDNA, and $X_i$ and $X_f$ correspond to the initial and final frame number. Velocities and distance travelled were plotted in Prism 6 as scatter plots and histograms which were fit to Gaussian distributions. Mean and SD were calculated from the distribution of the data. Significance was determined by P values (unpaired t tests).

## MMS spot assays

Wild type *SRS2*, along with 500 base pair upstream of the start codon, was amplified and cloned into yeast integrative vector, pRS303 (a gift from the R. Rothstein laboratory) using PCR-based method. Point mutations were obtained by site-directed mutagenesis. Vectors containing wildtype and point mutants of *SRS2* (full–length Srs2 without a GFP tag) were linearized at the *HIS3* locus using NheI restriction enzyme and transformed into yECG13 yeast strain (a gift from the R. Rothstein laboratory). Positive and single copy integrations into the host chromosome were confirmed by PCR and used for spot growth assays.

For spot assays, yeast strains were grown overnight in liquid YPD at 30 °C and diluted to OD₆₀₀ of 1.0 in water. Serially diluted cultures (4 µl) were spotted on freshly poured YPD and YPD plates containing 0.005% methyl methane sulfonate (MMS; Sigma-Aldrich; Cat. No. 129925). The plates were incubated at 30 °C and imaged 2 days post spotting.

## Spontaneous *LEU2* direct-repeat recombination assays

Spontaneous recombination rates were measured by employing the *leu2*-EcoRI::*URA3*::*leu2*-BsteII recombination system[79]. The recombination rate for each genotype was determined three times by performing fluctuation test on eight independent colonies belonging to each genotype. Strains were grown onto YPD plates for 3 days and independent single colonies of similar size were resuspended in 1 mL of water. Cells were then plated onto YPD and SC-Leu-Ura plates after appropriate dilutions (10⁴ for YPD and 10⁰ for SC-Leu-Ura) to determine the number of colony-forming units and the number of Leu⁺ recombinants, respectively. Colonies were counted after 3 days of incubation at 30 °C and the recombination rates calculated using the Lea and Coulson method of median[100].

## Crossover assays

Physical analysis of ectopic recombination was performed essentially as described[81]. In brief, cells were grown on YP–dextrose (2% glucose) or YP–galactose (2% galactose) plates for 2 days and Southern blot analysis of the digested genomic DNA performed to investigate the distribution of crossover and non–crossover products. Galactose was used for the induction of HO to generate double strand breaks. Repair efficiency was evaluated by testing the BamHI restriction enzyme sensitivity of the PCR amplified *ura3* gene using the flowing primers: 5'–AGA AAC ATG AAA TTG CCC AG–3' and 5'–TGT GAG TTT AGT ATA CAT GC–3'. For Southern blot analysis, 2 µg of genomic DNA was digested overnight with ApaLI and PvuII. DNA fragments were separated on an agarose gel (0.45% agarose, 1×Tris-Borate-EDTA, 0.3 µg/mL ethidium bromide) at a constant voltage of 1 V/cm for -20 h. After depurination, denaturation and neutralization of the gel, DNA was transferred overnight in 2× SSC to positively charged nylon membranes (GE Healthcare Amersham Hybond-N + ) and was then immobilized by ultraviolet cross-linking (1200 J). NCO and CO products were detected using a URA3 probe (primers Olea251b (5'-GGA AGA ACG AAG GAA GGA GC-3') and Olea252b (5'-TAA CGT TCA CCC TCT ACC TTA GC-3')) labelled by PCR amplification with ³²P dCTP (Perkin Elmer). ULTRA-hyb Ultrasensitive hybridization buffer (Invitrogen) was used to hybridize the probe overnight at 42 °C. Membranes were washed as recommended by the manufacturer. Blots were exposed in a phosphor screen cassette and the signal was detected with a Typhoon FLA 9000 (GE healthcare).

Image analysis and CO quantifications were done with ImageJ. For each strain and condition, the raw percent of CO was calculated as the ratio of CO bands from chromosomes V and II to the total amount of recombination products (all CO + NCO bands). CO quantifications were performed for at least three biological replicas for each strain.

## Statistical information

All statistical analysis was carried out using Graphpad Pris*m* Version 6.0. For all data error bars represent the standard deviation (SD) of the data. Statistical significance between groups was calculated using unpaired t-test. ns = not significant, *$p < 0.05$, **$p < 0.01$, ***$p < 0.001$, ****$p < 0.0001$.

## Statistical analysis

**In vitro biochemical experiments.** For the bulk biochemical ATP hydrolysis (Fig. 2a and Supplementary Tables 1, 2), DNA binding (Fig. 2b and Supplementary Table 1, 2), DNA helicase assays (Fig. 2c and Supplementary Table 1, 2) and D-loops disruption (Fig. 5b, d, f) the error bars represent SD calculated from three separate reactions. P values for the biochemical assays were calculated using a two-tailed Student's t–test.

**Single molecule DNA curtain experiments.** For the single molecule Srs2[898] translocation analysis and error bars represent 95% confidence intervals (CI). The number of single Srs2[898] molecules measured (N) are specified in each binding distribution histogram. For the single

molecule Srs2[898] velocity and processivity data error bars represent standard deviation (SD) and the N values are presented in each corresponding figure panel. The statistical parameters (velocity ± SD; processivity ± SD; N values; and corresponding figure panels) for all velocity and processivity measurements are summarized in Supplementary Tables 2 and 3. Note, that the N values for all single molecule experiments represent the number of single molecules that were analyzed for each given experiment and reflect the cumulative data collected from at least three separate flowcells. P values for the single molecule velocity and processivity assays (Fig. 3c–f, and Supplementary Tables 2 and 3) were calculated using a two-tailed Student's t-test.

**In vivo genetics assays.** The spontaneous recombination rate (Fig. 4d and Supplementary Fig. 4b) was determined from three independent fluctuation tests and the associated statistical significance was determined by unpaired t tests. For crossover (CO) assays, image analysis and CO quantifications were done with ImageJ. For each strain and condition (Fig. 6b, d and Supplementary Fig. 5b), the raw percent of CO was calculated as the ratio of CO bands from chromosomes V and II to the total amount of recombination products (all CO + NCO bands). CO quantifications were performed for at least three biological replicas for each strain. Error bars represent the standard deviation (SD) of the data. Statistical significance between groups was calculated using unpaired t-test.

### Reporting summary
Further information on research design is available in the Nature Portfolio Reporting Summary linked to this article.

## Data availability
All the information generated and analyzed is included in the manuscript and all graphs have associated raw data that is provided as an Excel worksheet as a Source Data file. The PDB entry 2IS1 was used for modelling. Source data are provided with this paper.

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

## Acknowledgements

We thank Rodney Rothstein and Luke Berchowitz for yeast strains and access to equipment. We thank members of the Greene and Symington laboratories for critically reading the manuscript. This research was funded by NIH grants R01CA236606 and R35GM118026 (E.C.G.), and R35GM126997 (L.S.S.). We thank Jerry Chang and the Precision Biomolecular Characterization Facility (PBCF) at Columbia University for technical support and access to the CD spectrometer, and we thank Sahiti Kuppa for assistance with the CD measurements. The PBCF is supported by NIH Award 1S10OD025102.

## Author contributions

A.M. cloned and purified all mutant proteins and conducted all single molecule assays and all bulk biochemical assays. V.B.R. constructed all yeast strains and conducted all genetic analyses of mutant proteins. C.E.R. assisted with the cloning of mutant protein expression constructs. L.M. and L.S.S. assisted with the design, implementation, and interpretation of the crossover assays. A.M., V.B.R and E.C.G. co–wrote the manuscript with input from all other co–authors.

## Competing interests

The authors declare no competing interests.
