## [Peer Review File · Nature Communications]

The separation pin distinguishes the pro- and anti-recombinogenic functions of *Saccharomyces cerevisiae* Srs2Reviewer #1 (Remarks to the Author):

This paper from the Greene and Symington laboratories addresses one of the long-standing mysteries in the study of non-replicative helicases/translocases. Helicases are traditionally appointed with DNA unwinding functions. Translocases move on DNA and remove protein roadblocks. Certain proteins possess robust translocation activity, but have been shown to unwind DNA (albeit slower) as they oligomerize on the DNA. The evidence and importance for such activities were built on decades on in vitro work on the UvrD system. Yet, the relative functional importance of translocation versus unwinding was never clear from an in vivo functional standpoint. Here, the prototypical example from yeast, Srs2, is used as a model to test the contributions of the DNA unwinding versus DNA translocation. The ssDNA translocation and associated Rad51 displacement activities of Srs2 are well established. Although a direct homolog of Srs2 in higher eukaryotes remains elusive, several candidates have been postulated i.e., functions done by Srs2 in yeast are likely split between a cadre of translocases in humans.

Here, using an AlphaFold model of Srs2, the team has identified a rather cool separation of function mutant of Srs2. The translocation, Rad51 clearing, DNA binding and ATPase activity of Srs2 are not affected, but the DNA unwinding activity is perturbed. While not readily evident, the characterization of these activities for a variety of mutants is a significant amount of work. The biochemical and single molecule data are paired with an exquisite array of in vivo characterization that brings home the point that the DNA unwinding activity is not required for HR and repair associated roles of Srs2.

I do not have many issues with the paper and the findings. Its rigorous, well-written, the data are clear and to the point, and each aspect of Srs2 activity is probed using multiple experimental approaches. This is a well-done study.

I do have a few minor points pertaining to the characterization of the DNA unwinding activity.

Minor Points:

What are the concentrations of DNA and Srs2 used in the unwinding experiments? These are not stated in the Methods or the Figure Legend. This is important as assembly of multiple Srs2 molecules are needed for DNA unwinding. Since the conditions are met for WT-Srs2 unwinding, I do not expect the findings to change. But, concentrations should be spelled out.

I am uncomfortable calling the truncated Srs2 protein as WT. Another notation should be used. Srs2t?

Can the authors perform a comparison of the WT and the pin-mutation in a full-length background? This is just to avoid any sort of weird allosteric affects.

It would be helpful to see a Coomassie stained and fluorescence image of an SDS-PAGE gel of the various mutant Srs2-GFP proteins used in the study.

There are a couple of papers that show that Srs2 unwinds DNA as a dimer. These need to be cited as they are directly relevant to the work presented here.

SI Figure 2c. Why does the unwinding reaction not increase as a function of time? i.e., the unwound strand should be increasing, but it's very inconsistent with a progressive unwinding reaction. This might be likely due to the FAM-DNA used in the experiment. Although its weird to see PIFE effects if the reaction is deproteinated. A better unwinding substrate will be a BHQ-Cy5 pair positioned at the blunt end of the substrate. Then follow the Cy5 fluorescence as Srs2 unwinds.

Reviewer #2 (Remarks to the Author):

In this manuscript, Meir *et al.* use an AlphaFold-generated structure of the Srs2 helicase, to rationally design mutants to identify residues that are important in ssDNA binding and helicase activity. They test these mutants for ATPase activity and ssDNA binding activity using gel-based assays. Using a single-molecule fluorescence DNA-curtains-based assay that is very well-established in the Greene lab, they test the mutants for translocation velocity and processivity on Rad51 and RPA coated ssDNA. Their results show that Srs2-Y775 is a separation-of-function mutant that can remove Rad51 and RPA from ssDNA, but is incapable of unwinding dsDNA and thus disrupting D-loops. Using this mutant in in *vivo* studies, they then determine that the main role of Srs2 is to remove Rad51 from ssDNA, while its dsDNA unwinding capability plays only a minor role.

The article is well structured, the experiments are carried out and analyzed in a rigorous fashion, and the data support the main conclusions. The findings will be of interest to the broader genomic maintenance field, specifically those interested in helicases, including human-disease related helicases, and homologous recombination.

I have the following minor comments:

1. The F285A and H650A mutants have a strong reduction in ATPase activity and ssDNA binding activity, and are inactive in all other assays. In the discussion, the authors highlight that these results could be consistent with loss of motor protein activity and loss of coupling during structural transitions. However, the results could also be consistent with the protein simply not being folded properly/stable enough. How important are these residues for 'proper' protein folding/stability. Can the authors show (possibly *in silico*) that these mutants fold correctly?
2. Pg 2: "...biochemical and biophysical characteristics can led to...". 'led' should read 'lead'.
3. On pg 4, when listing the mutants that exhibited large reductions in translocation velocity on Rad51-ssDNA, P671 is missing from the list, while it clearly shows reduction in Fig 3C.
4. On pg 4, the authors note that it is interesting that the R286A mutant displays moderate reduction in velocity and processivity on RPA-ssDNA. However, in the sentence before, they say that this reduction is not significant. If the authors want to keep this statement, it would be good to highlight that it is not significant. Alternatively, the authors would need to increase the number of data points to potentially reach significance.
5. Pg 5: "While... RPA-ssDNA." is an incomplete sentence.
6. Pg 6: "In these assays, ... previously described" is a confusing sentence.
7. Pg 7: "..., this data ..." should be "these data".
8. On pg 8 in the discussion, the authors reiterate that Y775 is highly conserved and say this strongly suggests that therefore it must be contributing to an important function otherwise it may not have been retained. I feel this is worded too strongly. Amino acids or functions are often retained if there is no selective pressure to remove them, even if the function is no longer on a molecular pathway that is used frequently.

Reviewer #3 (Remarks to the Author):

This manuscript by Meir et al. describes characterization of many mutants of the E. coli Sfla helicase Srs2. Despite there being no structures of Srs2, the authors are able to make well-informed mutations to investigate the protein's many roles including ATP hydrolysis, DNA binding, translocation and processivity, antirecombinase and hyper-recombination activity, and D-loop disruption. The results here demonstrate an impressive undertaking both in terms of the amount of mutants characterized and the breadth of experiments carried out. The authors provide compelling evidence that one of mutants, Y775A, is actually a separation-of-function mutant. Like WT, it is capable of removing Rad51, but unlike WT, it can't disrupt D-loops. In vivo experiments then lead the authors to the conclusion that the primary role of Srs2 during HR is removal of Rad51.

Overall, the manuscript is well written, and the results support the author's conclusions. I have a handful of major comments and some minor recommendations.

Major comments

1. For ATP hydrolysis, there should be a negative control without ssDNA since ATP-hydrolysis is

DNA-dependent.

2. Regarding the single molecule studies of Srs2 mutants:

a. Ideally, the full TIRFM videos should be made available. Would this be possible? Alternatively, the kymographs used for analysis should be main available as a resource for others who might want to reproduce the analysis conducted.

b. I have some confusion regarding the experimental setup and the explanatory figure panels 3a and 3b. Is the RPA-GFP tagged like it says in 3b and in the main text? I'm guessing that is a typo since the Srs2 is GFP labeled. The main text also mentions labeling with mCherry, which is not in the figure.

c. In the cartoons, it looks like the 3' end of the DNA is tethered, but for rolling circle amplification which intrinsically extends from 5' -> 3', the 5' end would be the biotin-tethered end. Srs2 moves in the 3' -> 5' direction since it is a Sf1a helicase. Thus, Srs2 should move against the flow from toward the tethered end of the DNA, which agrees with the kymographs, but isn't clear from the cartoons.

3. Figure 6d: This is not normalized, it is just scaled to the size of the bars from plot b. If it were normalized, it should be 100. I think the way the data are presented is a bit misleading, and it should be plotted as percent of CO and CO outcomes where the total is 100%. As it is, srs2Δ is plotted in such a way that it looks as though there are not any crossover events, when there clearly are a significant amount that can be seen in the gel in panel c.

Minor suggestions:

1. In the introduction, the Sf1 helicase family core is described in detail. A figure/cartoon would help clarify what is being described.

2. "This N-terminal GFP-Srs2 fusion construct retains biological function in vivo and retains biochemical activity in vitro." How much activity does it retain? Can you specify if it is fully active?

3. In the introduction, WT Srs2 is introduced as the GFP-Srs2 1-898 truncated, tagged variant. Because of this, it is unclear about what the controls in the in vivo experiments are. Are the controls the full-length, un-tagged WT? Perhaps indicating where full-length vs truncated WT is used by using FL or T, respectively, would be less ambiguous.

4. Please include panels for all mutants used in supplementary figure 2.

5. There are multiple instances of fluorescein being imaged with a phosphor imager. Instead mention what channels/wavelengths were used.

6. There are some references in the main text to figure 5a-5c which would be better suited to reference all of figure 5.

7. The first sentence in "Physical analysis of ectopic recombination in srs2-Y775A cells" introduces a lot of background not already mentioned. It would be nice to expand on it in a few sentences to provide clarity, especially for readers not well-versed in this particular kind of experiment. One example of this is also the first mention of the MAT locus, but to someone unfamiliar with this assay, it is unclear why this is mentioned.

8. "... perhaps because RPA has a much higher affinity for ssDNA compared to RAD51." Include reference.

9. Figure 1: While referring to an AlphaFold model as a homology model is correct, it is not as descriptive as simply calling it an AlphaFold model because a homology model could also be predicted with tools that tend to be less accurate. (a) What is the organism? Please indicate the important regions/residue general locations here such as (1) where DNA binds, (2) where the

residues in panel d are located, (3) where the pin domain is located, (4) where K41 is, (5) where the truncated 276 C-term residues are, (6) where ATP binds, etc. (b) It is not a "structure" but a "model" or "predicted structure". More information about the AlphaFold prediction would be very useful here such as coloring by pLDDT scores and including a legend, and perhaps including a PAE plot as well. (c) I think "merged structure" is misleading. Consider saying "aligned structures" instead. Make the cartoons more transparent in the overlay. (d) The floating residues are disorienting. Showing the backbone. Would provide significant clarity. I think that the figure might need to be re-worked to show all of the relevant information and include multiple views showing the key parts of the model.

10. Figure 2: (a, b) How many replicates were measured? Why aren't the legends consistent if the same proteins are being measured in both plots? It is impossible to distinguish many of the traces so it might be better to plot with thinner lines. Where is the data for Y775? "% Bound" is not a descriptive title for the y-axis. What is bound (ssDNA)? Also, would this plot and fits be better on a log scale? Make error bars thinner and black so that they can be seen. (c) normalized to what? Find a way to indicate on the plot itself what the three bars represent for each mutant.

11. Figure 3: The error bars are very difficult to see. Consider somehow making it more clear which half of the figure corresponds to RPA and which to Rad51. Maybe put a line separating the two halves and a large title.

12. Figure 4: (a) red/green arrows are hard to see/distinguish. Maybe switch to a chart format to show the different outcomes. (d) the bars that are light gray are really hard to see, especially when the paper is physically printed. It would be nice if the bars had the same colors used for the mutants in previous figures.

13. Figure 5: How many replicates? (a,c,e) label the concentration of protein used on these panels. In the legend, "total] minus Srs2" is confusing. I recommend removing "]" and changing "minus" to "-".

14. Figure 6: How many replicates? (d) y-axis title typo

15. SI Figure 1: see Figure 1 comments about merged vs. aligned structure. (a) Why are only some of the helicase motifs labeled, and why don't they match the names of the motifs given in the main text (Q, I, Ia, II, III, IV, V, VI vs. 1A, 1B, 2A, 2B). (b and c) these are unnecessary since they just provide the same information provided in main text figure 1. (d) While coloring by motif helps to see where the regions are in the structure, it would be more enlightening to see similarity clearly. Consider showing the aligned sequences colored by similarity, and perhaps boxing the motif regions. I know that only 2 of the aligned sequences are shown, but in the intro the similarity in that case of helicases is mentioned, so similarity coloring could be based on a full set of aligned sequences, even if only two of them are being shown in this figure.

Reviewer 1:

Reviewer summary: This paper from the Greene and Symington laboratories addresses one of the long-standing mysteries in the study of non-replicative helicases/translocases. Helicases are traditionally appointed with DNA unwinding functions. Translocases move on DNA and remove protein roadblocks. Certain proteins possess robust translocation activity but have been shown to unwind DNA (albeit slower) as they oligomerize on the DNA. The evidence and importance for such activities were built on decades on in vitro work on the UvrD system. Yet, the relative functional importance of translocation versus unwinding was never clear from an in vivo functional standpoint. Here, the prototypical example from yeast, Srs2, is used as a model to test the contributions of the DNA unwinding versus DNA translocation. The ssDNA translocation and associated Rad51 displacement activities of Srs2 are well established. Although a direct homolog of Srs2 in higher eukaryotes remains elusive, several candidates have been postulated i.e., functions done by Srs2 in yeast are likely split between a cadre of translocases in humans. Here, using an AlphaFold model of Srs2, the team has identified a rather cool separation of function mutant of Srs2. The translocation, Rad51 clearing, DNA binding and ATPase activity of Srs2 are not affected, but the DNA unwinding activity is perturbed. While not readily evident, the characterization of these activities for a variety of mutants is a significant amount of work. The biochemical and single molecule data are paired with an exquisite array of in vivo characterization that brings home the point that the DNA unwinding activity is not required for HR and repair associated roles of Srs2. I do not have many issues with the paper and the findings. Its rigorous, well-written, the data are clear and to the point, and each aspect of Srs2 activity is probed using multiple experimental approaches. This is a well-done study. I do have a few minor points pertaining to the characterization of the DNA unwinding activity.

Reviewer comment: What are the concentrations of DNA and Srs2 used in the unwinding experiments? These are not stated in the Methods or the Figure Legend. This is important as assembly of multiple Srs2 molecules are needed for DNA unwinding. Since the conditions are met for WT-Srs2 unwinding, I do not expect the findings to change. But, concentrations should be spelled out.

Response: 40 nM Srs2 was used for all helicase assays. This information has now been added to the methods and figure legends.

Reviewer comment: I am uncomfortable calling the truncated Srs2 protein as WT. Another notation should be used. Srs2t?

Response: We have changed the nomenclature and refer to the C-terminal truncated Srs2 protein as Srs2⁸⁹⁸. This nomenclature consistent with previous publications from our laboratory (Kaniecki et al, Cell reports, 2017).

Reviewer comment: Can the authors perform a comparison of the WT and the pin-mutation in a full-length background? This is just to avoid any sort of weird allosteric affects.

Response: Full-length Srs2 tends to aggregate in our hands, so the single-molecule experiments were conducted with a C-terminally truncated version of Srs2 comprised of amino acids 1 to 898 (Srs2⁸⁹⁸). The truncated Srs2 retains wild-type levels of ATPase, DNA helicase, and Rad51 filament disruption activities (please see Antony et al., 2009. Mol Cell 35: 105-115; Colavito et al., 2009. NAR 37: 6754-6764; Kaniecki et al., 2017. Cell Rep 21: 3166-3177).

Reviewer comment: It would be helpful to see a Coomassie stained and fluorescence image of an SDS-PAGE gel of the various mutant Srs2-GFP proteins used in the study.

Response: These gel images have now been added to Fig. S2a.

Reviewer comment: There are a couple of papers that show that Srs2 unwinds DNA as a dimer. These need to be cited as they are directly relevant to the work presented here.

Response: We apologize to the referee, but we are unaware of any studies demonstrating that Srs2 behaves like a dimer. We have shown that tandem arrays of Srs2 can load onto ssDNA during the removal of Rad51 (Kaniecki et al., 2017. Cell Rep 21: 3166-3177), and the Antony group has shown that there is a concentration-dependent phase during dsDNA unwinding, indicating that more one Srs2 molecule can participate (Lytle et al., 2014. JMB 426: 1883-1897); in both cases it seems clear that multiple Srs2 molecules are involved, but neither study claims that Srs2 forms a dimer. In addition, the Myong group has published data indicating that a monomer of Srs2 can undergo translocation and removal of Rad51 from ssDNA (Qiu, et al., 2013. Nat Commun 4: 2281). While we prefer to remain agnostic with respect to the potential oligomeric state of Srs2, if the referee has a specific reference in mind that conclusively demonstrates that Srs2 behaves as a dimer we would be more than happy to add it to the revision.

Reviewer comment: SI Figure 2c. Why does the unwinding reaction not increase as a function of time? i.e., the unwound strand should be increasing, but it's very inconsistent with a progressive unwinding reaction. This might be likely due to the FAM-DNA used in the experiment. Although it's weird to see PIFE effects if the reaction is deproteinated. A better unwinding substrate will be a BHQ-Cy5 pair positioned at the blunt end of the substrate. Then follow the Cy5 fluorescence as Srs2 unwinds.

Response: We thank the reviewer for this suggestion. We have now repeated all of the helicase assays using an Alexa-647 labeled substrate and the gels look much better. All of the helicase assay data in the revision has been replaced with the new data based upon the Alexa-647 assays.

Reviewer 2:

Reviewer summary: In this manuscript, Meir *et al.* use an Alphafold-generated structure of the Srs2 helicase, to rationally design mutants to identify residues that are important in ssDNA binding and helicase activity. They test these mutants for ATPase activity and

ssDNA binding activity using gel-based assays. Using a single-molecule fluorescence DNA-curtains-based assay that is very well-established in the Greene lab, they test the mutants for translocation velocity and processivity on Rad51 and RPA coated ssDNA. Their results show that Srs2-Y775 is a separation-of-function mutant that can remove Rad51 and RPA from ssDNA but is incapable of unwinding dsDNA and thus disrupting D-loops. Using this mutant in *in vivo* studies, they then determine that the main role of Srs2 is to remove Rad51 from ssDNA, while its dsDNA unwinding capability plays only a minor role.

The article is well structured, the experiments are carried out and analyzed in a rigorous fashion, and the data support the main conclusions. The findings will be of interest to the broader genomic maintenance field, specifically those interested in helicases, including human-disease related helicases, and homologous recombination.

I have the following minor comments:

Reviewer comment: The F285A and H650A mutants have a strong reduction in ATPase activity and ssDNA binding activity and are inactive in all other assays. In the discussion, the authors highlight that these results could be consistent with loss of motor protein activity and loss of coupling during structural transitions. However, the results could also be consistent with the protein simply not being folded properly/stable enough. How important are these residues for 'proper' protein folding/stability. Can the authors show (possibly *in silico*) that these mutants fold correctly?

Response: The strongest evidence we have that the F285A and H650A mutants are not unfolded are that they (1) purify similarly to Srs2⁸⁹⁸, (2) they remain fully soluble, and (3) these mutant proteins bind ssDNA with Kd values of 41 nM (F285A) and 46 nM (H650A; please see Supplementary Table 1). Although reduced compared to Srs2⁸⁹⁸ (Kd = 29 nM), Kd values in the range of tens of nanomolar still reflect very tight binding interactions, which would be inconsistent with protein misfolding. Lastly, rather than doing an *in-silico* measurement, we have opted to obtain experimental CD spectra for each of the mutant proteins used in this study. As is now shown in revised Supplementary Figs 3a & 3b, the CD spectra of the F285A and H650A mutants (and all of the other mutants) are comparable to Srs2⁸⁹⁸ and all of the proteins have similar alpha helical content, providing good evidence that the mutant proteins are properly folded.

Reviewer comment: Pg 2: "...biochemical and biophysical characteristics can led to...". 'led' should read 'lead'.

Response: Thank you for noting this error. It has been corrected in the revision.

Reviewer comment: On pg 4, when listing the mutants that exhibited large reductions in translocation velocity on Rad51-ssDNA, P671 is missing from the list, while it clearly shows reduction in Fig 3C.

Response: We apologize for this omission, the P671 mutant has now been added to the list.

Reviewer comment: On pg 4, the authors note that it is interesting that the R286A mutant displays moderate reduction in velocity and processivity on RPA-ssDNA. However, in the sentence before, they say that this reduction is not significant. If the authors want to keep this statement, it would be good to highlight that it is not significant. Alternatively, the authors would need to increase the number of data points to potentially reach significance.

Response: We have now rearranged the sentence emphasizing that although the moderate reduction is interesting, it is not statistically significant.

Reviewer comment: Pg 5: “While... RPA-ssDNA.” is an incomplete sentence.

Response: Thank you for noting this error, the incomplete sentence has now been corrected.

Reviewer comment: Pg 6: “In these assays, ... previously described” is a confusing sentence.

Response: We have rewritten the sentence to try to clarify its meaning.

Reviewer comment: Pg 7: “..., this data ...” should be “these data”.

Response: This error has now been corrected.

Reviewer comment: On pg 8 in the discussion, the authors reiterate that Y775 is highly conserved and say this strongly suggests that therefore it must be contributing to an important function otherwise it may not have been retained. I feel this is worded too strongly. Amino acids or functions are often retained if there is no selective pressure to remove them, even if the function is no longer on a molecular pathway that is used frequently.

Response: We agree with the reviewer. In the revision, we have deleted the sentence regarding the retention of this residue though evolution and instead simply speculate that it may be fulfilling an unknown role.

Reviewer 3:

Reviewer summary: This manuscript by Meir et al. describes characterization of many mutants of the E. coli Sf1a helicase Srs2. Despite there being no structures of Srs2, the authors are able to make well-informed mutations to investigate the protein’s many roles including ATP hydrolysis, DNA binding, translocation and processivity, antirecombinase and hyper-recombination activity, and D-loop disruption. The results here demonstrate an impressive undertaking both in terms of the number of mutants characterized and the breadth of experiments carried out. The authors provide compelling evidence that one of

mutants, Y775A, is actually a separation-of-function mutant. Like WT, it is capable of removing Rad51, but unlike WT, it can't disrupt D-loops. In vivo experiments then lead the authors to the conclusion that the primary role of Srs2 during HR is removal of Rad51.

Overall, the manuscript is well written, and the results support the author's conclusions. I have a handful of major comments and some minor recommendations.

Reviewer comment: For ATP hydrolysis, there should be a negative control without ssDNA since ATP-hydrolysis is DNA-dependent.

Response: The control experiment of Srs2 ATP hydrolysis without the presence of ssDNA has been added to the results. Srs2 shows no appreciable ATP hydrolysis activity when ssDNA is omitted from the reactions (please see Fig. 2a)

Reviewer comment: Regarding the single molecule studies of Srs2 mutants:

a. Ideally, the full TIRFM videos should be made available. Would this be possible? Alternatively, the kymographs used for analysis should be main available as a resource for others who might want to reproduce the analysis conducted.

Response: All kymographs used in the study will be provided to Nature Communications as raw data files as part of the final submission upon acceptance of the manuscript.

b. I have some confusion regarding the experimental setup and the explanatory figure panels 3a and 3b. Is the RPA-GFP tagged like it says in 3b and in the main text? I'm guessing that is a typo since the Srs2 is GFP labeled. The main text also mentions labeling with mCherry, which is not in the figure.

Response: We thank the reviewer for noting this error. The RPA used in both panels was tagged with mCherry and this information has been corrected in the text and in the figure.

c. In the cartoons, it looks like the 3' end of the DNA is tethered, but for rolling circle amplification which intrinsically extends from 5' -> 3', the 5' end would be the biotin-tethered end. Srs2 moves in the 3' -> 5' direction since it is a Sf1a helicase. Thus, Srs2 should move against the flow from toward the tethered end of the DNA, which agrees with the kymographs, but isn't clear from the cartoons.

Response: In these experiments, both ends of the DNA are tethered: the 5' end is tethered to the bilayer via a biotin-streptavidin linkage and the 3' end is anchored to the pedestals through non-specific surface absorption. This information has now been clarified in the main text, the methods section, and the figure panels, and we have also included a reference where we the use of this method for studying Srs2 in extensive detail (please see de Tullio et al., 2018. Single-stranded DNA curtains for studying the Srs2 helicase using total internal reflection fluorescence microscopy. *Methods in Enzymology* **600**, 407–437).

Reviewer comment: Figure 6d: This is not normalized, it is just scaled to the size of the bars from plot b. If it were normalized, it should be 100. I think the way the data are presented is a bit misleading, and it should be plotted as percent of CO and CO outcomes where the total is 100%. As it is, *srs2Δ* is plotted in such a way that it looks as though there are not any crossover events, when there clearly are a significant amount that can be seen in the gel in panel c.

Response: We thank the reviewer for noting this issue and we agree with the reviewer that the data is not normalized to the plating efficiency. We have now corrected this point in the main text, figure, and figure legend.

With regards to the presentation of data, for these experiments an equal amount of DNA is loaded onto the gel for Southern blot analysis. However, there is a difference in the number of cells that are able to grow after the induction of galactose. Since most of the *srs2* delete cells do not grow upon induction of galactose cannot be sure what is the percentage of CO/NCO in those cells. Presenting the data as percent of CO and NCO with total being 100% will neglect this information and we believe it can be misleading to readers. We understand that there are a significant number of CO events observed for *srs2* in the gel and it can be confusing to relate the CO's observed in gel with the quantification of the scaled up plating efficiency of the same and that is why we have provided the raw values pertaining to the percent of CO's in all the strains in both the main text and as a table in Supplementary figure 4b.

Minor suggestions:

Reviewer comment: In the introduction, the Sf1 helicase family core is described in detail. A figure/cartoon would help clarify what is being described.

Response: We have recently published a review paper which discusses in depth the structure of Sf1a helicases (Meir et. al., Genes, 2021); this and other papers are cited. We have also modified Fig. 1 to help clarify the structural details (see below).

Reviewer comment: "This N-terminal GFP-Srs2 fusion construct retains biological function *in vivo* and retains biochemical activity *in vitro*." How much activity does it retain? Can you specify if it is fully active?

Response: We have cited the original research articles describing the retention of both *in vivo* and *in vitro* biological functions of N-terminally GFP-Srs2 constructs. The Rothstein group showed that YFP-tagged Srs2 was functional *in vivo* by investigating the viability with either *rad54Δ* or *sgs1Δ* alleles, both of which render *srs2* mutant strains inviable (please see Burgess et al., 2009. J Cell Biol. 185: 969-981). In the same study, this YFP-Srs2 construct was also shown to co-localize with Rad52 at DNA repair foci. We have also previously shown that GFP-tagged versus untagged Srs2 have comparable levels of ATP

hydrolysis activity; and the translocation velocities and translocation distances for labeled and unlabeled Srs2 proteins on Rad51-ssDNA are statistically indistinguishable (Kaniecki et al., 2017. Cell Rep 21: 3166-3177).

Reviewer comment: In the introduction, WT Srs2 is introduced as the GFP-Srs2 1-898 truncated, tagged variant. Because of this, it is unclear about what the controls in the *in vivo* experiments are. Are the controls the full-length, un-tagged WT? Perhaps indicating where full-length vs truncated WT is used by using FL or T, respectively, would be less ambiguous.

Response: Full-length untagged or tagged Srs2 is prone to aggregation in our hands, which is why we perform experiments with the Srs2 1-898 variant. To clarify this issue, we have now changed the nomenclature of truncated GFP-Srs2 1-898 protein from WT-Srs2 to Srs2⁸⁹⁸ throughout the entire manuscript. All of the *in vitro* assays are performed with Srs2⁸⁹⁸ whereas all the *in vivo* assays utilized full-length untagged Srs2.

Reviewer comment: Please include panels for all mutants used in supplementary figure 2.

Response: Supplementary figure 2 is a combination of three different experiments with each experiment conducted with all 11 mutants – in the figure we have chosen to show WT Srs2 (now labeled Srs2⁸⁹⁸) and either the H650A mutant (panels b and c) or the Y775A mutant (panel d). We chose to show the H650A mutant in the ATPase assay (panel b) because it was the most distinct from Srs2⁸⁹⁸; we chose to show the H650A mutant in the gel shift assay (panel c) to emphasize the point that although this mutant has highly compromised ATP hydrolysis activity, it can still bind fairly tightly to ssDNA; we chose to show the Y775A mutant in the helicase assay (panel d) to emphasize that it does not unwind DNA. Given the number of mutants analyzed, if we were to show all of them in this figure it would require 33 panels. As an alternative, we will instead provide all the panels used for these experiments as separate raw images that we will be submitting to the journal prior to final acceptance.

Reviewer comment: There are multiple instances of fluorescein being imaged with a phosphor imager. Instead mention what channels/wavelengths were used.

Response: We have incorporated this suggestion in the revised methods section. In brief, the images were scanned at 473 nm wavelength using a Typhoon FLA 9000 phosphor imager (GE Healthcare).

Reviewer comment: There are some references in the main text to figure 5a-5c which would be better suited to reference all of figure 5.

Response: Thank you for noting these errors and they have been corrected in the revision.

Reviewer comment: The first sentence in “Physical analysis of ectopic recombination in srs2-Y775A cells” introduces a lot of background not already mentioned. It would be nice

to expand on it in a few sentences to provide clarity, especially for readers not well-versed in this particular kind of experiment. One example of this is also the first mention of the MAT locus, but to someone unfamiliar with this assay, it is unclear why this is mentioned.

Response: We have expanded this section and provided more details for clarification.

Reviewer comment: "... perhaps because RPA has a much higher affinity for ssDNA compared to RAD51." Include reference.

Response: We have now included the following references indicating that RPA acts as a kinetic barrier to Rad51 filament formation: Sugiyama et al., 1997. JBC 272: 7940-7945; Sung et al., 1997. Genes Dev 11: 1111-1121; and Kowalczykowski, 2015. Cold Spring Harb Perspect Biol. 7: a016410.

Reviewer comment: Figure 1: While referring to an AlphaFold model as a homology model is correct, it is not as descriptive as simply calling it an AlphaFold model because a homology model could also be predicted with tools that tend to be less accurate. (a) What is the organism? Please indicate the important regions/residue general locations here such as (1) where DNA binds, (2) where the residues in panel d are located, (3) where the pin domain is located, (4) where K41 is, (5) where the truncated 276 C-term residues are, (6) where ATP binds, etc. (b) It is not a "structure" but a "model" or "predicted structure". More information about the AlphaFold prediction would be very useful here such as coloring by pLDDT scores and including a legend, and perhaps including a PAE plot as well. (c) I think "merged structure" is misleading. Consider saying "aligned structures" instead. Make the cartoons more transparent in the overlay. (d) The floating residues are disorienting. Showing the backbone. Would provide significant clarity. I think that the figure might need to be re-worked to show all of the relevant information and include multiple views showing the key parts of the model.

Response: We now refer to the structural model as an "AlphaFold model". (a) The organism is *S. cerevisiae* as is now indicated in the first paragraph of the results section as well as the methods section. We changed the colors to magenta (Srs2) and green (UvrD), so each structure/model is more noticeable in the overlay (Figure 1c). The separation pin, bound ATP and bound DNA are now shown in the structure of UvrD (Figure 1a). We have added the AlphaFold model of the truncated Srs2⁸⁹⁸, modeled in the structure of the DNA (taken from the UvrD structure), and further highlighted the specific residues mutated in this study with color-coded space filling residues (Figure 1d). The AlphaFold model of Srs2 was adapted from the open access database where all the information such as pLDDT and PAE plots are publicly accessible.

Reviewer comment: Figure 2: (a, b) How many replicates were measured? Why aren't the legends consistent if the same proteins are being measured in both plots? It is impossible to distinguish many of the traces so it might be better to plot with thinner lines. Where is the data for Y775? "% Bound" is not a descriptive title for the y-axis. What is bound (ssDNA)?

Also, would this plot and fits be better on a log scale? Make error bars thinner and black so that they can be seen. (c) normalized to what? Find a way to indicate on the plot itself what the three bars represent for each mutant.

Response: The number of replicates, three in each case, has been added to the figure legend. The plots have been made with thinner lines. We acknowledge that there is a lot of data which can make the different mutants difficult to distinguish, and for this reason the quantitative values obtained from each plot are presented in Supplementary Tables 1 and 2 to assist the reader in interpreting the behaviors of each mutant protein. Please also note that when publishing a paper in Nature Communications we are asked to include an excel file that contains all of the individual data points (and all of the error bar values) presented in all of the figures throughout the entire manuscript. Interested readers will be able to refer to this spread sheet to see all of the exact data point values for every experiment in the manuscript. We have now changed the Y-axis label in panel b to “% ssDNA bound”. Changing the error bars to all black makes the figure more difficult to interpret, so we have kept the original color-coded scheme; note that all of these error bar values will be provided to Nature Communications as a spread sheet to accompany the final accepted version of the manuscript. In panel c, the data are normalized to the maximum unwound product seen for Srs2⁸⁹⁸ and we have now indicated within the figure itself that the three bars for each protein tested represent 5-, 15-, and 30-minute time points. Panels made with either log or linear scales (see below) look comparable, so we have kept our original linear scale panel in the main figures, but we will be happy switch if the referee prefers the log scale panel.

Reviewer comment: Figure 3: The error bars are very difficult to see. Consider somehow making it more clear which half of the figure corresponds to RPA and which to Rad51. Maybe put a line separating the two halves and a large title.

Response: We have increased the line widths for the error bars and have now placed labels above each panel to clarify which correspond to RPA and which correspond to Rad51. In addition, we will be submitting all of the raw data values and corresponding error values for all graphs in the manuscript as an excel spreadsheet upon final acceptance.

Reviewer comment: Figure 4: (a) red/green arrows are hard to see/distinguish. Maybe switch to a chart format to show the different outcomes. (d) the bars that are light gray are really hard to see, especially when the paper is physically printed. It would be nice if the bars had the same colors used for the mutants in previous figures.

Response: We have changed the colors of the arrows and the bars as suggested by the referee.

Reviewer comment: Figure 5: How many replicates? (a,c,e) label the concentration of protein used on these panels. In the legend, “total] minus Srs2” is confusing. I recommend removing “]” and changing “minus” to “-“.

Response: The experiments were repeated in triplicate, and this point is now noted in the figure legend. We have chosen to keep the “]” to remain consistent with the other data sets presented in the legend, but we have changed “minus” to “-“.

Reviewer comment: Figure 6: How many replicates? (d) y-axis title typo

Response: The number of biological replicates (three in each instance) used was mentioned in the methods section of the original submission. In the revision, we have now also added this information in the figure legend. Thank you for pointing out the y-axis typo; it has now been corrected.

Reviewer comment: SI Figure 1: see Figure 1 comments about merged vs. aligned structure. (a) Why are only some of the helicase motifs labeled, and why don't they match the names of the motifs given in the main text (Q, I, Ia, II, III, IV, V, VI vs. 1A, 1B, 2A, 2B). (b and c) these are unnecessary since they just provide the same information provided in main text figure 1. (d) While coloring by motif helps to see where the regions are in the structure, it would be more enlightening to see similarity clearly. Consider showing the aligned sequences colored by similarity, and perhaps boxing the motif regions. I know that only 2 of the aligned sequences are shown, but in the intro the similarity in that case of helicases is mentioned, so similarity coloring could be based on a full set of aligned sequences, even if only two of them are being shown in this figure.

Response: There seems to be some confusion between the protein domains (1A, 1B, 2A, 2B) and the helicase motifs (Q, I, Ia, II, III, IV, V, VI), which we have tried to clarify in the main text, and we also cite several references explaining these in further detail. We do not name any of the motifs in Figure 1, instead we only highlight the domains of the proteins (domains 1A, 2A, 1B and 2B) and all of these domains are mentioned in the main text along with relevant citations. In Supplemental Figure 1, we then show all of the motifs (Q, I, Ia, II, III, IV, V, VI), highlighting their locations within domains 1A, 2A, 1B, and 2B (Supplemental Figure 1a-c) and we also show their locations relative to the primary structure of the protein (Supplemental Figure 1d). We used the same color coding in the 3D structures and models

(Supplemental Figure 1a-c) and the primary sequences (Supplemental Figure 1d) to help orient the reader regarding their locations. In our view, adding further color coding to Supplemental Figure 1d would make the figure more confusing.

Reviewer #1 (Remarks to the Author):

The authors have addressed all of my concerns. Again, wonderful study!

Reviewer #2 (Remarks to the Author):

The authors have addressed my comments. The additional CD data that were added in Supplementary Fig. 3 are compelling evidence that all proteins are properly folded.

In my opinion, this manuscript is now acceptable for publication.

Reviewer #3 (Remarks to the Author):

I appreciate the authors responses to my comments and efforts to address them. I remain convinced that the experiments and data are of high significance and should be published. However, in several places the authors claimed to have addressed my comments, but no changes were actually made. Other comments were only partially addressed. Therefore, I still believe the manuscript needs improvement prior to publication.

The following comments were not fully addressed.

a. Ideally, the full TIRFM videos should be made available. Would this be possible? Alternatively, the kymographs used for analysis should be made available as a resource for others who might want to reproduce the analysis conducted.

Author Response: All kymographs used in the study will be provided to Nature Communications as raw data files as part of the final submission upon acceptance of the manuscript.

My Response: I am not sure what "raw data files" for kymographs is. Is that the full TRIFM videos that others could look at?

c. In the cartoons, it looks like the 3' end of the DNA is tethered, but for rolling circle amplification which intrinsically extends from 5' -> 3', the 5' end would be the biotin-tethered end. Srs2 moves in the 3' -> 5' direction since it is a Sf1a helicase. Thus, Srs2 should move against the flow from toward the tethered end of the DNA, which agrees with the kymographs, but isn't clear from the cartoons.

Author Response: In these experiments, both ends of the DNA are tethered: the 5' end is tethered to the bilayer via a biotin-streptavidin linkage and the 3' end is anchored to the pedestals through non-specific surface absorption. This information has now been clarified in the main text, the methods section, and the figure panels, and we have also included a reference where we the use of this method for studying Srs2 in extensive detail (please see de Tullio et al., 2018. Single-stranded DNA curtains for studying the Srs2 helicase using total internal reflection fluorescence microscopy. *Methods in Enzymology* 600, 407–437).

My Response: The response states that the information has been clarified in multiple places including the methods section. I am not sure where in the methods section it has been clarified.

Reviewer comment: Figure 6d: This is not normalized, it is just scaled to the size of the bars from plot b. If it were normalized, it should be 100. I think the way the data are presented is a bit misleading, and it should be plotted as percent of CO and CO outcomes where the total is 100%. As it is, srs2Δ is plotted in such a way that it looks as though there are not any crossover events, when there clearly are a significant amount that can be seen in the gel in panel c.

Author Response: We thank the reviewer for noting this issue and we agree with the reviewer that the data is not normalized to the plating efficiency. We have now corrected this point in the main text, figure, and figure legend. With regards to the presentation of data, for these experiments an equal amount of DNA is loaded onto the gel for Southern blot analysis. However, there is a

difference in the number of cells that are able to grow after the induction of galactose. Since most of the *srs2* delete cells do not grow upon induction of galactose cannot be sure what is the percentage of CO/NCO in those cells. Presenting the data as percent of CO and NCO with total being 100% will neglect this information and we believe it can be misleading to readers. We understand that there are a significant number of CO events observed for *srs2* in the gel and it can be confusing to relate the CO's observed in gel with the quantification of the scaled up plating efficiency of the same and that is why we have provided the raw values pertaining to the percent of CO's in all the strains in both the main text and as a table in Supplementary figure 4b.

My Response: I disagree. If the authors insist on presenting the data as such, then I recommend adding an additional panel that shows the NCO and CO events in a normalized fashion.

Reviewer comment: "This N-terminal GFP-Srs2 fusion construct retains biological function in vivo and retains biochemical activity in vitro." How much activity does it retain? Can you specify if it is fully active?

Author Response: We have cited the original research articles describing the retention of both in vivo and in vitro biological functions of N-terminally GFP-Srs2 constructs. The Rothstein group showed that YFP-tagged Srs2 was functional in vivo by investigating the viability with either *rad54Δ* or *sgs1Δ* alleles, both of which render *srs2* mutant strains inviable (please see Burgess et al., 2009. *J Cell Biol.* 185: 969-981). In the same study, this YFP-Srs2 construct was also shown to co-localize with Rad52 at DNA repair foci. We have also previously shown that GFP-tagged versus untagged Srs2 have comparable levels of ATP hydrolysis activity; and the translocation velocities and translocation distances for labeled and unlabeled Srs2 proteins on Rad51-ssDNA are statistically indistinguishable (Kaniecki et al., 2017. *Cell Rep* 21: 3166-3177).

My Response: This response does not address my question. Citing that it retains functionality does not describe how functional it is in relation to the full-length Srs2.

Reviewer comment: Please include panels for all mutants used in supplementary figure 2.

Author Response: Supplementary figure 2 is a combination of three different experiments with each experiment conducted with all 11 mutants – in the figure we have chosen show WT Srs2 (now labeled Srs2898) and either the H650A mutant (panels b and c) or the Y775A mutant (panel d). We chose to show the H650A mutant in the ATPase assay (panel b) because it was the most distinct from Srs2898; we chose to show the H650A mutant in the gel shift assay (panel c) to emphasize the point that although this mutant has highly compromised ATP hydrolysis activity, it can still bind fairly tightly to ssDNA; we chose to show the Y775A mutant in the helicase assay (panel d) to emphasize that it does not unwind DNA. Given the number of mutants analyzed, if we were to show all of them in this figure it would require 33 panels. As an alternative, we will instead provide all the panels used for these experiments as separate raw images that we will be submitting to the journal prior to final acceptance.

My Response: All information should be provided in the SI as panels, even if you have to make additional panels. It should be easily accessible to the reader, not just as raw images.

Reviewer comment: There are multiple instances of fluorescein being imaged with a phosphor imager. Instead mention what channels/wavelengths were used.

Author Response: We have incorporated this suggestion in the revised methods section. In brief, the images were scanned at 473 nm wavelength using a Typhoon FLA 9000 phosphor imager (GE Healthcare).

My Response: The fact that you used a Typhoon capable of phosphor imaging does not mean that you used phosphor imaging. This is incorrect. It's like saying that using a color printer to print a black and white document is the same as printing the document in color.

Reviewer comment: The first sentence in "Physical analysis of ectopic recombination in *srs2*-Y775A cells" introduces a lot of background not already mentioned. It would be nice to expand on it in a

few sentences to provide clarity, especially for readers not well-versed in this particular kind of experiment. One example of this is also the first mention of the MAT locus, but to someone unfamiliar with this assay, it is unclear why this is mentioned.

Author Response: We have expanded this section and provided more details for clarification.

My Response: I cannot find where this section was changed.

Reviewer comment: Figure 1: While referring to an AlphaFold model as a homology model is correct, it is not as descriptive as simply calling it an AlphaFold model because a homology model could also be predicted with tools that tend to be less accurate. (a) What is the organism? Please indicate the important regions/residue general locations here such as (1) where DNA binds, (2) where the residues in panel d are located, (3) where the pin domain is located, (4) where K41 is, (5) where the truncated 276 C-term residues are, (6) where ATP binds, etc. (b) It is not a "structure" but a "model" or "predicted structure". More information about the AlphaFold prediction would be very useful here such as coloring by pLDDT scores and including a legend, and perhaps including a PAE plot as well. (c) I think "merged structure" is misleading. Consider saying "aligned structures" instead. Make the cartoons more transparent in the overlay. (d) The floating residues are disorienting. Showing the backbone. Would provide significant clarity. I think that the figure might need to be re-worked to show all of the relevant information and include multiple views showing the key parts of the model.

Author Response: We now refer to the structural model as an "AlphaFold model". (a) The organism is *S. cerevisiae* as is now indicated in the first paragraph of the results section as well as the methods section. We changed the colors to magenta (Srs2) and green (UvrD), so each structure/model is more noticeable in the overlay (Figure 1c). The separation pin, bound ATP and bound DNA are now shown in the structure of UvrD (Figure 1a). We have added the AlphaFold model of the truncated Srs2898, modeled in the structure of the DNA (taken from the UvrD structure), and further highlighted the specific residues mutated in this study with color-coded space filling residues (Figure 1d). The AlphaFold model of Srs2 was adapted from the open access database where all the information such as pLDDT and PAE plots are publicly accessible.

My Response: "The AlphaFold model of Srs2 was adapted from the open access database where all the information such as pLDDT and PAE plots are publicly accessible." This is a vital part of your manuscript. It should be included in the SI so that readers can easily see it without looking it up in the database.

Reviewer #1 (Remarks to the Author):

The authors have addressed all of my concerns. Again, wonderful study!

New Author Response: Thank you.

Reviewer #2 (Remarks to the Author):

The authors have addressed my comments. The additional CD data that were added in Supplementary Fig. 3 are compelling evidence that all proteins are properly folded.

In my opinion, this manuscript is now acceptable for publication.

New Author Response: Thank you.

Reviewer #3 (Remarks to the Author):

I appreciate the authors responses to my comments and efforts to address them. I remain convinced that the experiments and data are of high significance and should be published. However, in several places the authors claimed to have addressed my comments, but no changes were actually made. Other comments were only partially addressed. Therefore, I still believe the manuscript needs improvement prior to publication.

The following comments were not fully addressed.

a. Ideally, the full TIRFM videos should be made available. Would this be possible? Alternatively, the kymographs used for analysis should be made available as a resource for others who might want to reproduce the analysis conducted.

Original Author Response: All kymographs used in the study will be provided to Nature Communications as raw data files as part of the final submission upon acceptance of the manuscript.

My Response: I am not sure what “raw data files” for kymographs is. Is that the full TRIFM videos that others could look at?

New Author Response: No, the videos are not provided. All data analysis was performed using kymographs, not videos. All the kymographs used for the entire study (698 kymographs in total) are being provided for inspection by any interested readers.

c. In the cartoons, it looks like the 3' end of the DNA is tethered, but for rolling circle amplification which intrinsically extends from 5' -> 3', the 5' end would be the biotin-tethered end. Srs2 moves in the 3' -> 5' direction since it is a Sfla helicase. Thus, Srs2 should move against the flow from toward the tethered end of the DNA, which agrees with the kymographs, but isn't clear from the cartoons.

Original Author Response: In these experiments, both ends of the DNA are tethered: the 5' end is tethered to the bilayer via a biotin-streptavidin linkage and the 3' end is anchored to the pedestals through non-specific surface absorption. This information has now been clarified in the main text, the methods section, and the figure panels, and we have also included a reference where we the use of this method for studying Srs2 in extensive detail (please see de Tullio et al., 2018. Single-stranded DNA curtains for studying the Srs2 helicase using total internal reflection fluorescence microscopy. Methods

in *Enzymology* 600, 407–437).

My Response: The response states that the information has been clarified in multiple places including the methods section. I am not sure where in the methods section it has been clarified.

New Author Response: The methods used for the DNA curtain experiments are fully described in the methods section entitled “**DNA curtain assays and data analysis**”. In the main text under the subtitle “**Single molecule studies of Srs2⁸⁹⁸ mutants**” there is also a long introduction which describes how these experiments are performed, explains the orientations of the ssDNA, and includes several references to prior publications, including a reference to a *Methods in Enzymology* paper which explains everything in great detail with respect to the DNA curtain studies of Srs2. We are reproducing the relevant main text from the revised manuscript below:

“Srs2 can actively translocate along ssDNA while stripping both RPA and Rad51 from the ssDNA^{40,41,43}, and we have established single molecule ssDNA curtains assays allowing for the direct observation of GFP-tagged Srs2⁸⁹⁸ in real time using total internal reflection fluorescence microscopy (TIRFM)^{47,48,76}. In brief, long ssDNA substrates (≥ 50 kilonucleotides, knts) are generated by rolling circle replication using a 5' biotin-labeled ssDNA primer and the resulting 5' biotinylated ssDNA is tethered to a supported lipid bilayer on the surface of a microfluidic sample chamber through a biotin-streptavidin linkage⁷⁶. The ssDNA molecules are then aligned at chromium (Cr) nanofabricated barriers to lipid diffusion, which are deposited onto the fused silica by electron beam lithography⁷⁶. Addition of mCherry-labeled RPA allows the ssDNA to be extended by hydrodynamic force, the 3' ends of the RPA-ssDNA become anchored to Cr pedestals through nonspecific adsorption, allowing the molecules to be visualized by TIRFM (Fig. 3a & 3b)⁷⁶. Once assembled, the RPA can be displaced by the addition of Rad51 plus ATP resulting in the formation of long Rad51-ssDNA filaments⁷⁶. Using these types of assays, we have previously reported that GFP-Srs2⁸⁹⁸ translocates at a rate of 142 ± 77 nucleotides per second (nts/sec) for an average distance of 18.5 ± 0.65 kilonucleotides (knt) on ssDNA that is bound by Rad51 and 170 ± 80 nt/sec for an average distance of 14.4 ± 0.40 knt on RPA bound ssDNA^{47,48}.”

In addition, we have now modified the figure panels to include labeled arrowheads indicating the “*Direction of Srs2 movement*” (please see below). And, we have amended the accompanying figure legend to include the following text “*Note that Srs2 moves in the 3' to 5' direction on the ssDNA as indicated by the arrows in the schematic diagrams.*”

Reviewer comment: Figure 6d: This is not normalized, it is just scaled to the size of the bars from plot b. If it were normalized, it should be 100. I think the way the data are presented is a bit misleading, and it should be plotted as percent of CO and NCO outcomes where the total is 100%. As it is, *srs2Δ* is plotted in such a way that it looks as though there are not any crossover events, when there clearly are a significant amount that can be seen in the gel in panel c.

Original Author Response: We thank the reviewer for noting this issue and we agree with the reviewer that the data is not normalized to the plating efficiency. We have now corrected this point in the main text, figure, and figure legend. With regards to the presentation of data, for these experiments an equal amount of DNA is loaded onto the gel for Southern blot analysis. However, there is a difference in the number of cells that are able to grow after the induction of galactose. Since most of the *srs2* delete cells do not grow upon induction of galactose cannot be sure what is the percentage of CO/NCO in those cells. Presenting the data as percent of CO and NCO with total being 100% will neglect this information and we believe it can be misleading to readers. We understand that there are a significant number of CO events observed for *srs2* in the gel and it can be confusing to relate the CO's observed in gel with the quantification of the scaled up plating efficiency of the same and that is why we have provided the raw values pertaining to the percent of CO's in all the strains in both the main text and as a table in Supplementary figure 5b.

My Response: I disagree. If the authors insist on presenting the data as such, then I recommend adding an additional panel that shows the NCO and CO events in a normalized fashion.

New Author Response: We have added the requested panel of normalized to NCO and CO events to Supplementary Figure 5c.

Reviewer comment: "This N-terminal GFP-Srs2 fusion construct retains biological function in vivo and retains biochemical activity in vitro." How much activity does it retain? Can you specify if it is fully active?

Original Author Response: We have cited the original research articles describing the retention of both in vivo and in vitro biological functions of N-terminally GFP-Srs2 constructs. The Rothstein group showed that YFP-tagged Srs2 was functional in vivo by investigating the viability with either *rad54Δ* or *sgs1Δ* alleles, both of which render *srs2* mutant strains inviable (please see Burgess et al., 2009. J Cell Biol. 185: 969-981). In the same study, this YFP-Srs2 construct was also shown to co-localize with Rad52 at DNA repair foci. We have also previously shown that GFP-tagged versus untagged Srs2 have comparable levels of ATP hydrolysis activity; and the translocation velocities and translocation distances for labeled and unlabeled Srs2 proteins on Rad51-ssDNA are statistically indistinguishable (Kaniecki et al., 2017. Cell Rep 21: 3166-3177).

My Response: This response does not address my question. Citing that it retains functionality does not describe how functional it is in relation to the full-length Srs2.

New Author Response: No, we cannot specify if it is fully active, which is why we more accurately state "...it retains biological function in vivo and retains biochemical activity in vitro". As indicated above, the YFP-tagged version of full length Srs2 is active in vivo based upon the cited genetic studies. As shown in a previous publication (Kaniecki et al., 2017. Cell Rep 21: 3166-3177), GFP-tagged Srs2⁸⁹⁸ and untagged Srs2⁸⁹⁸ have comparable levels of ATP hydrolysis activity (Kaniecki et al., 2017, please see Figure S1C); and the translocation velocities and translocation distances for labeled and unlabeled Srs2⁸⁹⁸ proteins on Rad51-ssDNA are statistically indistinguishable (Kaniecki et al., 2017,

please see Figure 1). The ATP hydrolysis activity of GFP-tagged Srs2⁸⁹⁸ and untagged Srs2⁸⁹⁸ are both ~2-fold higher than that of full length Srs2 (Kaniecki et al., 2017, please see Figure S1C), which we attribute to the strong propensity of full length Srs2 to form insoluble, inactive aggregates in solution. The propensity of full length Srs2 to aggregate was also previously reported by Antony, E., et al., 2009. Mol Cell 35: 105-115, as was the use of the Srs2⁸⁹⁸ truncation for overcoming the aggregation problem.

Reviewer comment: Please include panels for all mutants used in supplementary figure 2.

Original Author Response: Supplementary figure 2 is a combination of three different experiments with each experiment conducted with all 11 mutants – in the figure we have chosen show WT Srs2 (now labeled Srs2898) and either the H650A mutant (panels b and c) or the Y775A mutant (panel d). We chose to show the H650A mutant in the ATPase assay (panel b) because it was the most distinct from Srs2898; we chose to show the H650A mutant in the gel shift assay (panel c) to emphasize the point that although this mutant has highly compromised ATP hydrolysis activity, it can still bind fairly tightly to ssDNA; we chose to show the Y775A mutant in the helicase assay (panel d) to emphasize that it does not unwind DNA. Given the number of mutants analyzed, if we were to show all of them in this figure it would require 33 panels. As an alternative, we will instead provide all the panels used for these experiments as separate raw images that we will be submitting to the journal prior to final acceptance.

My Response: All information should be provided in the SI as panels, even if you have to make additional panels. It should be easily accessible to the reader, not just as raw images.

New Author Response: We are providing gel images for all of the experiments as part of the excel spreadsheet that is required for publication in Nature Communications. These gel images are all fully annotated and easily accessible to the reader.

Reviewer comment: There are multiple instances of fluorescein being imaged with a phosphor imager. Instead mention what channels/wavelengths were used.

Original Author Response: We have incorporated this suggestion in the revised methods section. In brief, the images were scanned at 473 nm wavelength using a Typhoon FLA 9000 phosphor imager (GE Healthcare).

My Response: The fact that you used a Typhoon capable of phosphor imaging does not mean that you used phosphor imaging. This is incorrect. It's like saying that using a color printer to print a black and white document is the same as printing the document in color.

New Author Response: The language has been changed to “Typhoon FLA 9000”.

Reviewer comment: The first sentence in “Physical analysis of ectopic recombination in srs2-Y775A cells” introduces a lot of background not already mentioned. It would be nice to expand on it in a few sentences to provide clarity, especially for readers not well-versed in this particular kind of experiment. One example of this is also the first mention of the MAT locus, but to someone unfamiliar with this assay, it is unclear why this is mentioned.

Original Author Response: We have expanded this section and provided more details for clarification.

My Response: I cannot find where this section was changed.

New Author Response: The following text was added to the manuscript in response to the reviewer's comment:

“HO endonuclease is responsible for initiating gene conversion at the mating type (MAT) locus by inducing DSBs. To circumvent this issue, the MATa site on Ch III is modified to MATa-inc allele which is refractory to HO digest and thus prevents cleavage of the MAT locus.”

This text can be found in the first paragraph (sixth sentence) under the Results section subheading **“Physical analysis of ectopic recombination in srs2-Y775A cells”**.

Reviewer comment: Figure 1: While referring to an AlphaFold model as a homology model is correct, it is not as descriptive as simply calling it an AlphaFold model because a homology model could also be predicted with tools that tend to be less accurate. (a) What is the organism? Please indicate the important regions/residue general locations here such as (1) where DNA binds, (2) where the residues in panel d are located, (3) where the pin domain is located, (4) where K41 is, (5) where the truncated 276 C-term residues are, (6) where ATP binds, etc. (b) It is not a “structure” but a “model” or “predicted structure”. More information about the AlphaFold prediction would be very useful here such as coloring by pLDDT scores and including a legend, and perhaps including a PAE plot as well. (c) I think “merged structure” is misleading. Consider saying “aligned structures” instead. Make the cartoons more transparent in the overlay. (d) The floating residues are disorienting. Showing the backbone. Would provide significant clarity. I think that the figure might need to be re-worked to show all of the relevant information and include multiple views showing the key parts of the model.

Original Author Response: We now refer to the structural model as an “Alphafold model”. (a) The organism is *S. cerevisiae* as is now indicated in the first paragraph of the results section as well as the methods section. We changed the colors to magenta (Srs2) and green (UvrD), so each structure/model is more noticeable in the overlay (Figure 1c). The separation pin, bound ATP and bound DNA are now shown in the structure of UvrD (Figure 1a). We have added the AlphaFold model of the truncated Srs2⁸⁹⁸, modeled in the structure of the DNA (taken from the UvrD structure), and further highlighted the specific residues mutated in this study with color-coded space filling residues (Figure 1d). The AlphaFold model of Srs2 was adapted from the open access database where all the information such as pLDDT and PAE plots are publicly accessible.

My Response: “The AlphaFold model of Srs2 was adapted from the open access database where all the information such as pLDDT and PAE plots are publicly accessible.” This is a vital part of your manuscript. It should be included in the SI so that readers can easily see it without looking it up in the database.

New Author Response: We have added the pLDDT and PAE plots to the SI. Please see Supplemental Figure 1d & 1e.